# Prokaryotic viruses impact functional microorganisms in nutrient removal and carbon cycle in wastewater treatment plants

Yiqiang Chen[1], Yulin Wang [1], David Paez-Espino [2], Martin F. Polz [3,4] & Tong Zhang [1✉]

As one of the largest biotechnological applications, activated sludge (AS) systems in wastewater treatment plants (WWTPs) harbor enormous viruses, with 10-1,000-fold higher concentrations than in natural environments. However, the compositional variation and host-connections of AS viruses remain poorly explored. Here, we report a catalogue of ~50,000 prokaryotic viruses from six WWTPs, increasing the number of described viral species of AS by 23-fold, and showing the very high viral diversity which is largely unknown (98.4-99.6% of total viral contigs). Most viral genera are represented in more than one AS system with 53 identified across all. Viral infection widely spans 8 archaeal and 58 bacterial phyla, linking viruses with aerobic/anaerobic heterotrophs, and other functional microorganisms controlling nitrogen/phosphorous removal. Notably, Mycobacterium, notorious for causing AS foaming, is associated with 402 viral genera. Our findings expand the current AS virus catalogue and provide reference for the phage treatment to control undesired microorganisms in WWTPs.

[1] Environmental Microbiome Engineering and Biotechnology Laboratory, Center for Environmental Engineering Research, Department of Civil Engineering, The University of Hong Kong, Hong Kong, China. [2] Department of Energy, Joint Genome Institute, Lawrence Berkeley National Laboratory, Berkeley, CA, USA. [3] Department of Civil and Environmental Engineering, Massachusetts Institute of Technology, Cambridge, MA, USA. [4] Division of Microbial Ecology, Centre for Microbiology and Environmental Systems Science, University of Vienna, Vienna, Austria. ✉email: zhangt@hku.hk

As one of the largest biotechnological applications on earth[1], the activated sludge (AS) system in wastewater treatment plants (WWTPs) is the embodiment of artificially controlling the activity of microbial communities to perform services to humans. The AS system contains enormous microbial diversity, including ~1 billion bacterial phylotypes worldwide[2] that form high biomass (2–50 g/L) in condensed flocs or granules. Viruses are an important but poorly explored component of this system, reaching 10–1000-fold higher concentrations than their counterparts in other aquatic systems[3,4]. Through specific interactions such as host lysis or lysogeny, viruses can directly impact prokaryotic communities and have been implicated in causing 41% of the variation in community composition in anaerobic digesters (AD), considerably >15% estimated due to abiotic factors[5]. Considering this potential importance of viruses in AS systems, we asked what the diversity and variation are among different WWTPs and how viruses may impact prokaryotic taxa, especially those that contribute to the removal of organic matter and inorganic nutrients.

With the rapid progress of metagenomics, the discovery of viral sequences in ecosystems has been enormously accelerated. Paez-Espino et al.[6] identified 125k partial sequences of DNA viruses from 3042 global metagenomic data sets, expanding the known viral gene pool by 16-fold. Roux et al. uncovered >15k viral populations from ocean metagenomic data, revealing the fundamental ecological functions they performed in global biogeochemical networks[7]. Recently, the diversity of huge phages (bacterial viruses)[8] and giant viruses[9] were also investigated at the global scale through metagenomic data, making it a powerful tool to decipher viral diversity in ecosystems. We, therefore, reasoned that metagenomics is a powerful discovery tool to unravel the diversity of DNA viruses in WWTPs.

To identify and analyze the diversity of prokaryotic viruses in AS systems, we applied metagenomic sequencing to DNA virus-enriched samples from six biological WWTPs in Hong Kong. We amassed a catalog of ~50,000 prokaryotic viruses, which expands the current AS virome database (IMG/VR database v.2.0[10]) by 12-fold at the genus level and by 23-fold at the species level. We discovered substantial uncharacterized viral diversity in AS communities, with only between 0.4% and 1.6% of total viral contigs (coverage percentage) being assigned to a known viral family. Most viral genera were represented in more than one AS system and 53 identified across all samples constituted a common virome. By predicting hosts, we show that viruses may impact functional microorganisms in biological nutrient removal and the carbon cycle in WWTPs, linking viruses with aerobic/anaerobic heterotrophs, and other functional microorganisms controlling N/P removal. For example, the AS foaming bacteria Mycobacterium (~3% in the AS microbial community) was associated with 402 viral genera. This work thus provides a genomic reference for the potential future design of the phage treatment to tackle sludge foaming problems.

## Results

### AS systems display many novel and a high fraction of shared viruses.
We used metagenomic sequencing of 30 Gb of sequences per sample to characterize the composition of viral concentrates across six WWTPs (see Methods). After assembly and mapping of reads, 24–34% of the total sequence information could be classified as viral using two different identification pipelines (see Methods). By combining the results of these two pipelines, the final data set consisted of 50,037 viral contigs with an N50 >20 kb. To evaluate whether our sequencing effort sufficiently sampled the viromes, all six WWTP samples were subsampled iteratively to evaluate the saturation dynamics. Rarefaction curves of the

number of viral contigs, reached a plateau ~15 Gb of sequencing data for all six samples (Fig. 1a), indicating adequate recovery of prokaryotic viruses in these AS systems at the sequencing depth (30 Gb per sample) in this study.

Viral contigs were further classified into 8756 viral clusters (VC, equivalent to viral genera) using vConTACT2 by calculating the gene-content-based distance between viral contigs (~40% proteome similarity)[11], and each VC was assigned an ID for identification (mean length = 15.2 kb, mean genera size = 3). Compared with the current number of viral sequences from AS systems, our sequencing data increase the AS virome database ($N = 2103$ in the IMG/VR database v.2.0)[10] by 12-fold at the genus level and by 23-fold at the species-level (95% identity, 80% coverage). Comparison with NCBI RefSeq viral genome database showed that across the six AS systems, only 0.4–1.6% of total viral contigs (coverage percentage) could be assigned to a known viral family. Similar to previously described viral metagenomes from the soil, freshwater, and marine system[7,12,13], this limited annotation highlights substantial uncharacterized viral diversity in AS communities. Among these recognizable viruses, members of the family Podoviridae (short-tailed phages from the Caudovirales order) were the most prevalent, comprising on average 41.3% of these viral contigs (coverage percentage) across the six WWTPs.

All samples displayed high but variable diversity of viral genera with Shannon's diversity index H' ranging from 5.22 to 7.14 and Pielou's evenness index J' ranging from 0.71 to 0.86 (Supplementary Data 1). These differences are evident in rank-abundance curves, which show that each sample has different viral frequency patterns (Fig. 1b). Most viruses occurred at low frequency with the relative abundance of individual genera diminishing below 0.1% after counting the top 138 viral genera.

Principal coordinate analysis (PCoA) of the Bray–Curtis dissimilarity based on the relative abundance of viral genera suggested that most samples are divergent from each other (Fig. 1c), with only two pairs of AS viromes, ST and STL as well as SK and SWH, displaying higher similarity to each other.

The overall variability in the viromes is also reflected in different dominance patterns. Each AS sample yielded a different dominant viral genus, and while these were also abundant in some WWTPs, they were below the detection limit in others (Fig. 1d). Although high relative abundance across all WWTPs indicates linkage to consistently abundant hosts, highly variable occurrence suggests that host populations are also more dynamic.

Although the viromes appear overall variable in rank abundance, many viral genera were shared across the WWTPs. Fifty-three viral genera were detected in all samples and were thus considered to be common members of the AS viromes, accounting for 1.7–5.4% of viral contigs (coverage percentage) in each WWTP (Fig. 2). Thirteen of these common viral genera were also present in AS viromes in the IMG/VR database v.2.0[10]. Of the total of 8756 unique viral genera collected across samples, STL and ST contained the largest fraction (5245 and 5149 genera, respectively) and shared most viral genera ($N = 2885$) with each other, far exceeding the number of all the other shared or unique viral genera (Fig. 2). These two WWTPs also had only 45 and 64 site-specific viral genera, consistent with the pattern in the PCoA virome profile. On the other hand, SWH possessed the most unique viral genera ($N = 323$), making up 11.0% of the total (Fig. 2). In fact, only relatively few viral genera were found exclusively in one of the WWTPs.

Overall, these results suggest that the virome across WWTPs consists of many shared genera. The lack of detection of some viral genera in the AS virome of one WWTP may be primarily due to the biological variation in the grab samples and/or the technical variation. Hence, if such technical and biological

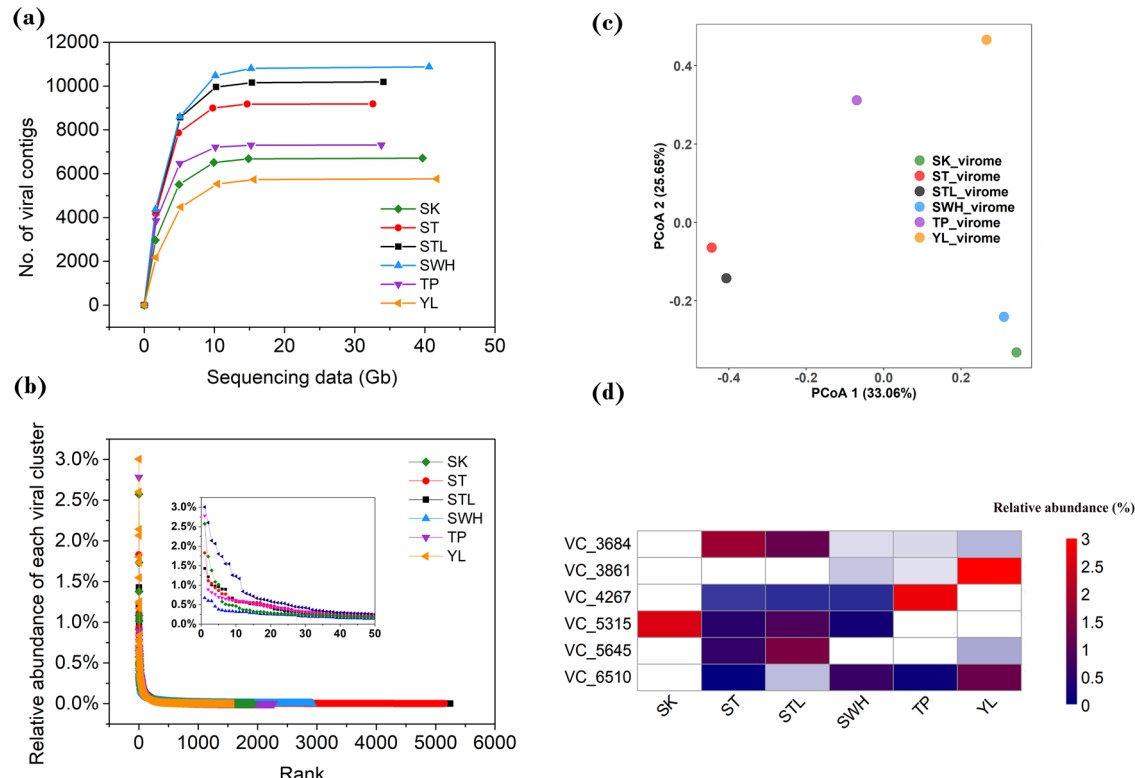

**Fig. 1 Compositional variation in viromes among wastewater treatment plants. a** Rarefaction curve of each sample. **b** Rank-abundance curve of each sample. **c** PCoA analysis of viromes based on the relative abundance of viral genera. **d** Relative abundance and appearance of dominant viral genera in each WWTP. White color denotes no appearance in this WWTP. Source data are provided as a Source Data file.

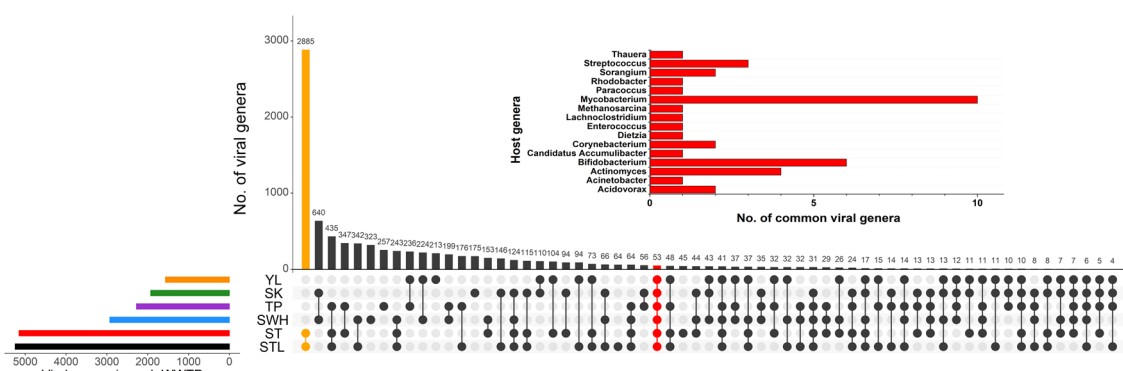

**Fig. 2 Shared viral genera in each WWTP.** The UpSet[55] chart shows the total number of viral genera and their sharedness in each WWTP. The bar chart on the top right shows the distribution of predicted hosts for common viral genera in all WWTPs. Shared viral genera between ST and STL were labeled orange and shared viral genera in all WWTPs were labeled red. Source data are provided as a Source Data file.

variations are taken into account, the virome shared among all AS maybe even more diverse.

**Viruses infect a broad spectrum of bacteria and archaea.** To examine putative host associations of all 50,037 viral contigs in the six WWTPs, we amassed a database of approximately three million CRISPR-Cas spacers from the NCBI prokaryotic complete genomes and metagenomes database (https://www.ncbi.nlm.nih.gov/assembly/). Host prediction was performed by matching CRISPR-Cas spacers at a sequence identity above 97%, sequence coverage over 90% in length, and mismatches <1 using the BLASTn-short task function (see Methods). Because the spacer content in a CRISPR array reflects recent virus encounters of bacteria and

archaea, this approach has the potential to accurately link viruses to their hosts[14] although it is obviously limited to hosts containing CRISPR-Cas systems[15]. Based on the cutoffs used, we recovered a total of 5879 viral contigs (4897 viral genera) (11.7% recall rate) with their predicted hosts, comprising 11.0–22.6% of viral contigs (coverage percentage) in each WWTP. Considering the random recall rate (0.70%) simply happens by chance (see Methods), the percentage of erroneous associations can be 6%.

At the phylum level, viruses were predicted to infect a wide range of archaea and bacteria, including 8 archaeal and 58 bacterial phyla (Fig. 3). Within the Archaea domain, 206 viruses were linked to Euryarchaeota, 10-fold more than to any other archaeal phylum and consistent with a previous survey of AS communities in WWTPs finding that Euryarchaeota was the

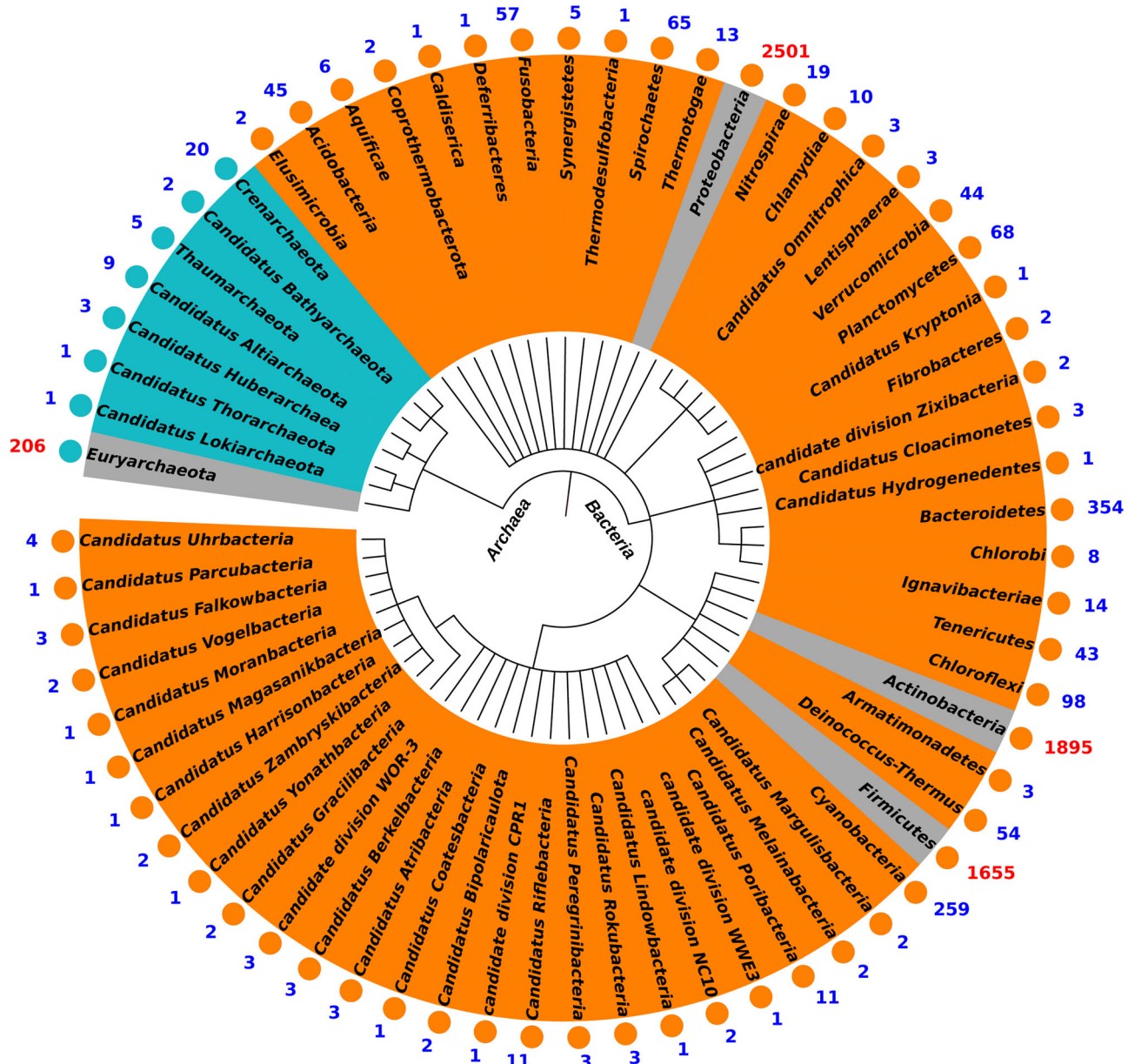

**Fig. 3 Taxonomy of predicted hosts of viral contigs from all samples at phylum level.** Eight phyla of archaea and 58 phyla of bacteria were shown in the graph. Green dots represented archaea and orange dots represented bacteria. The top three bacteria and top one archaea that could be infected by most viruses were highlighted with a gray background. Source data are provided as a Source Data file.

most abundant lineage of archaea[16]. Euryarchaeota-infecting viruses can account for 0.25–0.56% in the viral community. Among bacteria, viruses infecting Proteobacteria ($N = 2501$), Actinobacteria ($N = 1895$), and Firmicutes ($N = 1655$) were the most abundant. The relative abundance of these viruses is quite similar to each other, i.e., 6.06%, 5.53%, and 4.20% for Proteobacteria, Actinobacteria, and Firmicutes-infecting phages, respectively. The higher proportion of actinophages indicates that actinophages, which have recently been discovered to be the most dominant viral groups in freshwater[13,17], are also abundant in AS systems.

Out of the 53 common viral genera, 16 could be assigned to more narrowly defined putative host taxa (Fig. 2). These hosts included members of the aerobic heterotrophs Mycobacterium, Bifidobacterium, and Actinomyces, which play important roles in organic carbon removal in WWTPs, and were predicted to be infected by ten, six, and four common viral genera, respectively.

The methanogen Methanosarcina and a bacterium implicated in phosphorus removal, Candidatus Accumulibacter, also had predicted interactions with the common viral genera. These predicted interactions between common viral genera and functional microorganisms highlight the potential impact of viruses on the operational performance of WWTPs.

Matching of CRISPR spacers also suggested a range of host specificities of the AS viruses. The majority (68%) of CRISPR-annotated viral contigs (coverage percentage) were identified to be specialists at the host genus level, suggesting a narrow host range. However, 1668 out of 5879 viruses may be able to infect more than one host genus as suggested by the occurrence of CRISPR signatures in divergent host genomes. Intriguingly, 135 viruses might be able to infect hosts from different domains of life, with 73 displaying perfect matches to spacers in the CRISPR database, indicating viruses might occasionally switch hosts from bacteria to archaea. Considering the random error percentage of 6% in host identifications in our

samples, the possibility that each one of these identifications is false equals $P(f) = 1 - (1 - 0.06)^2 = 0.1164 = 11.6\%$. Therefore, of 135 viruses infecting two domains of life, ~16 (11.6%) would be expected to have happened just by chance. Whether such switching happens primarily on ecological or evolutionary timescales through existing broad-host affinity or recombination of receptor domains remains poorly understood. However, recent experimental and high-resolution host mapping has highlighted that broad-host-range viruses, including those able to infect different domains of life, are widely distributed in nature[18–20].

**Viruses may impact key functional microorganisms in WWTPs.** To further investigate the connections between functional microorganisms and affiliated viruses in WWTPs, putative host genera identified through CRISPR-Cas spacer matches were linked to metabolic functions according to their classification in the MIDAS database[21]. This data set was supplemented with manually selected methanogens and sulfate/sulfur-reducing microorganisms. An ensemble of 100 functional microorganisms at the genera level was recovered with links to 1345 viruses, accounting for an average of 25.1% of CRISPR-annotated viral contigs (coverage percentage) across the WWTPs. For those broad-host viruses, we assumed that they could indirectly impact multiple functions in WWTPs. For example, viral contig STL1812_19787 could infect both Candidatus Accumulibacter (phosphate-accumulating organism) and Desulfosoma (sulfate-reducing organism). By infecting these two hosts, STL1812_19787 could indirectly impact both the phosphate removal process and the sulfate-reducing process. In this case, we add one to the count of both Candidatus Accumulibacter and Desulfosoma in Fig. 4.

Virus-host connections were identified for 48 genera of aerobic heterotrophic bacteria (Fig. 4a). Seventeen of these genera were found to have virus–host connections in all the six WWTPs. Among them, Mycobacterium, Streptococcus, and Acinetobacter had the highest number of predicted viral interactions, i.e., 402, 263, and 109, respectively. Notably, Mycobacterium is notorious

for being the major sludge foaming bacteria in AS, accounting for an average of 3% of 16S rRNA sequences in a previous study about foaming bacteria dynamics over 5 years[22]. Extensive links between Mycobacterium and its corresponding viruses provide a valuable genomic database for the potential future design of phage treatment to tackle sludge foaming problems.

Our data (Fig. 4b) suggest that viruses can infect 53 functional anaerobic genera in AS, including 30 fermenters, 6 acetogens, and 17 methanogens. Anaerobic processes could occur in micro-environments within sludge flocs[23] in spite of aeration overall creating an unfavorable environment for anaerobes in the bulk water and outer layers of AS flocs. However, some viruses that display specificity for sulfate reducers (owing to seawater flushing in Hong Kong) and methanogens might be detected in the AS system due to their presence in the influent. Results showed that they account for a very low percentage (0.13–0.28%) in total viral contigs (coverage percentage). These viruses might be carried from the wastewater influent.

Viruses may also impact bacteria involved in nutrient removal from AS as indicated by virus-host connections to 47 bacterial genera previously implicated in nitrogen and phosphorus removal, and sulfur cycle (Fig. 4c). These include one genus of ammonia-oxidizing bacteria (AOB) and three genera of nitrite-oxidizing bacteria (NOB). Ten viral genera were found in five WWTPs that could infect the AOB Nitrosomonas. Polyphosphate-accumulating bacteria were represented by Candidatus Accumulibacter and Tetrasphaera, where the first had a total of 24 viral matches across all WWTPs. Finally, a wide range of sulfate-reducing bacteria, which can generate hydrogen sulfide from sulfate, were represented by 21 genera, where 16 and 6 viral genera had connections in all WWTPs with Desulfovibrio and Desulfosoma, respectively. These results suggest that viruses could indirectly impact biological nutrient removal and carbon cycling in WWTPs through lysis of key functional microorganisms in AS.

We have also checked the number of spacers in all Midas genera (functional microorganisms in WWTPs) in our study.

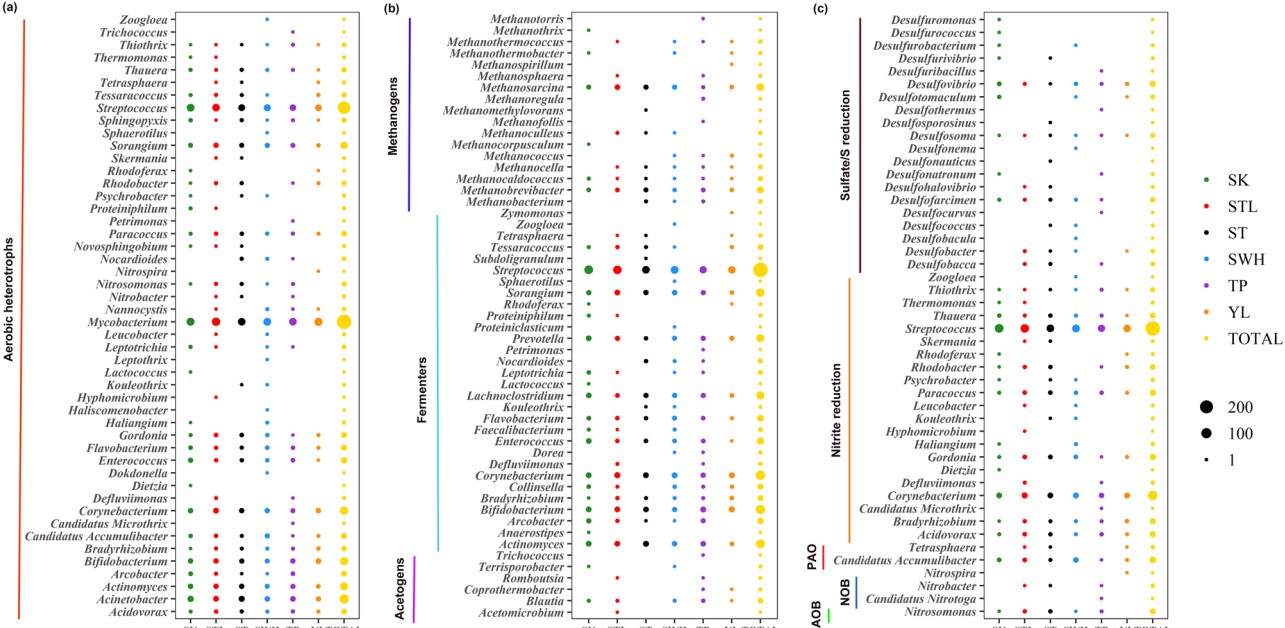

**Fig. 4 Virus–host profiles of functional microorganisms in WWTPs. a** Virus–host profiles of functional microorganisms related to aerobic oxidation. **b** Virus–host profiles of functional microorganisms related to the anaerobic process. **c** Virus–host profiles of functional microorganisms related to nutrient removal (N, P, S). The size of the circle indicates the number of viral genera that could infect the corresponding microbial genera. Source data are provided as a Source Data file.

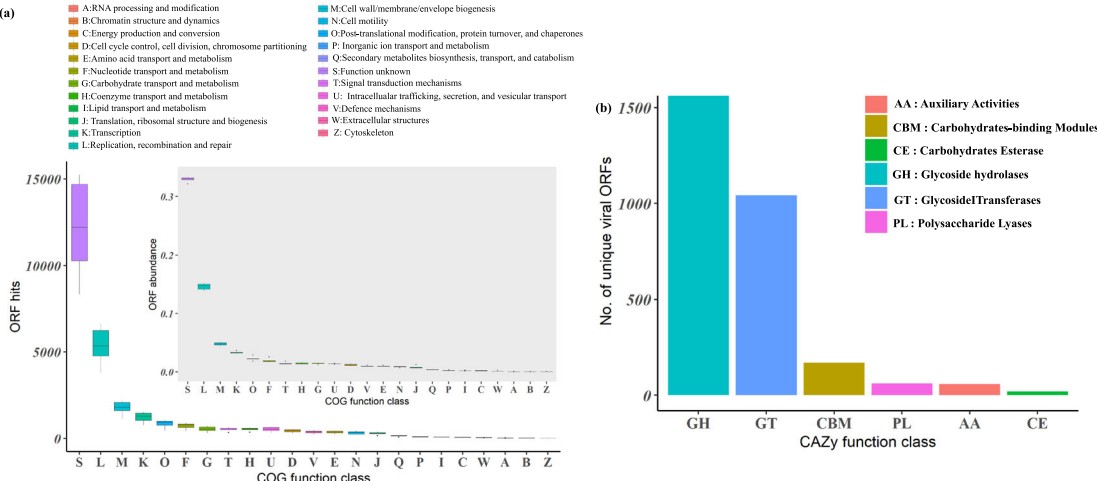

**Fig. 5 Distribution of auxiliary metabolic genes (AMGs) relevant to the carbon cycle and nutrient removal. a** Boxplot of the overall gene profile for six WWTPs was summarized both as viral ORF hits and viral ORF relative abundance (ORF hits to each COG function class/total ORFs that have hits to eggNOG database). Data in **a** are presented as mean values (center) and 25%, 75% percentiles (bounds of box). The minima and maxima represent the range of the data. **b** Number of unique viral ORFs related to each CAZy function class was shown in a descendant order. Source data are provided as a Source Data file.

Results showed that these Midas genera contain on average 44 spacers in their genome and there exists an uneven distribution of CRISPR in prokaryotic groups (Supplementary Data 8). It should be noticed that the number of phage-prey interactions that can be detected using this approach is a minority and does not always represent present scenarios. Even phages infecting a CRISPR-positive strain may leave no trace in the form of spacers. Therefore, the conclusions that derive from these predictions must be handled with caution.

**Viruses encode extensive auxiliary metabolic genes.** Viral ORFs were distributed across 23 COG functional classes, demonstrating their high diversity. Aside from those with unknown or hypothetical functions, on average >1000 ORFs in each sample were found in the following categories, L: replication, recombination, and repair, M: cell wall/membrane/envelope biogenesis, and K: transcription, confirming that the main functions sustain viral reproduction and transcription. It is noteworthy that viruses encode on average 541 ORFs (1.4% of annotated viral ORFs) in each sample in category G: carbohydrate transport and metabolism. After removing redundant viral ORFs, most unique ORFs ($N = 1610$) were classified into the glycoside hydrolases (GH) module[24] (Fig. 5b). These GHs may be involved in the digestion of capsules to allow the viral tail to reach its membrane receptor on the host. The high representation of this function may be explained by the prevalence of biofilm formation among AS microbes and has previously also been noted among mangrove sediment viruses[25].

Previous work has shown that many prokaryotic viruses carry auxiliary metabolic genes (AMGs), which can modulate host energy metabolism to provide an energetic advantage during viral genome and protein synthesis[26]. Broadly speaking, AMGs refer to all metabolic genes in lytic phages[27], i.e., all genes in categories C, E, F, G, H, I, and P. Though only about a quarter of the viral ORFs have annotations in the eggNOG database, our data reveal a large repertoire of potential AMGs in the AS viromes (Fig. 5a). For example, 72 ORFs in total are potentially involved with carbon fixation pathways and 35 of them annotate as photosynthetic carbon fixation pathways in KEGG[28]. Moreover, 10 viral ORFs belong to *CobS* genes, which are essential for the biosynthesis of cobalamin. Cobalamin biosynthesis pathway

is usually not complete in bacterial genomes[29], and these viral encoded *CobS* genes could possibly assist the host metabolic capability. Seven viral genes encode adenylyl-sulfate kinase (*CysC*), which could facilitate host's assimilatory sulfate reduction. By modulating host metabolism during infection, AMGs could alter the specific functions of their hosts in WWTPs and therefore influence carbon cycling and the removal of nutrients.

**Viromes are shared between WWTPs and the water environment.** To investigate whether the WWTP viral genera also occur in other habitats, we compared all viral sequences recovered here with the IMG/VR database v.2.0, which consists of 735,112 viral contigs predicted from metagenomic data[10]. Viral sequences from five ecosystems (AS, AD, solid waste, freshwater, marine) were included for comparison. For AS ($N = 2103$), AD ($N = 8580$), and solid waste ($N = 5760$), all viral sequences were subjected to our viral clustering pipeline, whereas for freshwater and marine environments, we each randomly selected 50,037 viral sequences to match the number in our samples ($N = 50,037$).

Results showed that viral genera in our samples were shared among multiple environments. There was considerable overlap between WWTPs and freshwater ($N = 402$) and marine ($N = 172$) environments (Fig. 6). Moreover, 200, 273, and 244 viral genera were shared with AS, AD, and solid waste, respectively (Fig. 6). This represents a higher number than with marine viromes, and if normalized against the dataset size, there is also a larger fraction of connections than with freshwater viromes. When hosts were predicted for these shared viral genera, Proteobacteria was the most shared host phylum, followed by Firmicutes, Actinobacteria, Bacteroidetes, and Cyanobacteria (Supplementary Data 2). At the genus level, Bacillus was the most abundant between our samples and marine, AD, and AS environments, while Streptomyces displayed a higher prevalence between our samples and freshwater and solid waste environments (Supplementary Data 3). Although it is difficult to identify the source and sink dynamics, AS and AD are typical processes in WWTPs, and solid waste viral sequences mainly stem from compost and leachate microbial communities. The considerable sharing of viromes between WWTPs and marine water may be caused by Hong Kong's extensive application of marine water for

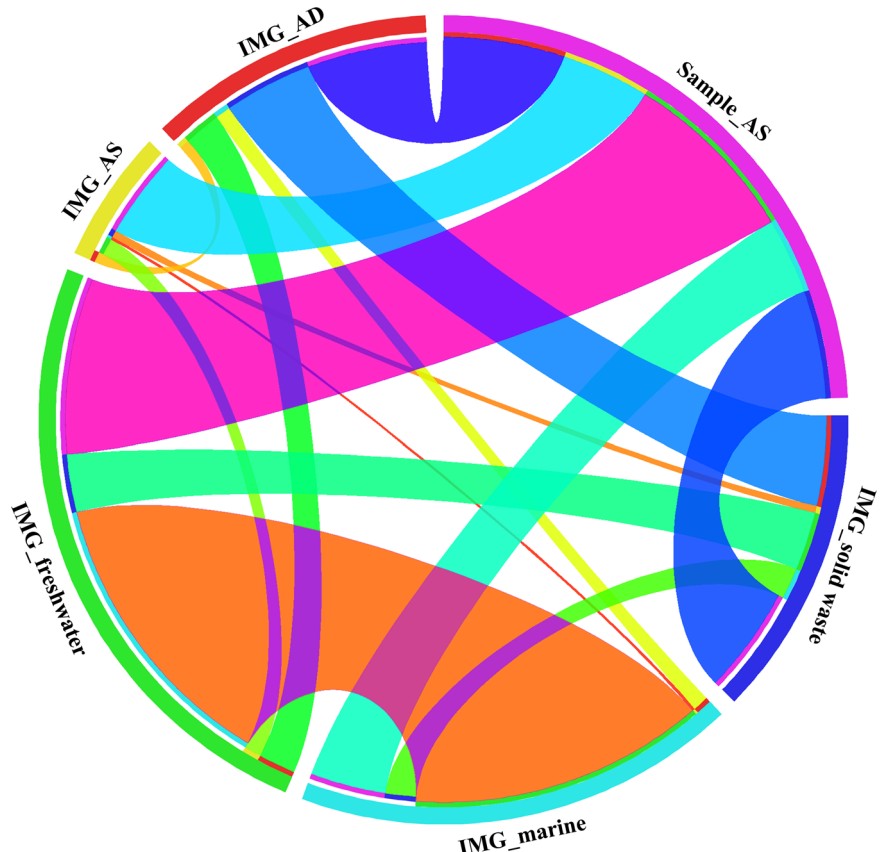

**Fig. 6 Connections between viral genera in AS samples and in five ecosystems from IMG/VR database.** Pairwise connections were shown in a Circos[56] plot between samples and different ecosystems, including AS, AD, solid waste, freshwater, and marine water. Source data are provided as a Source Data file.

toilet flushing, causing the influent sewage of WWTPs to contain a sizable amount of marine water. These data thus suggest that viruses are extensively shared and that the same viral genera may manipulate microbial communities in these different environments.

**Hi-C validation of virus–host interactions in AS system**. High throughput chromosome conformation capture (Hi-C) method was used to validate the virus–host connections predicted by our CRISPR-based methods using an additional sample in December 2020 at ST WWTP, by referring to viral contigs and host genome bins obtained from direct sequencing using Illumina and Nanopore metagenomic sequencing (Fig. 7).

As for Illumina metagenomic sequencing, 4578 viral contigs were identified and 1695 of them were deconvoluted in Hi-C data to have virus–host interactions with 197 host bins (Supplementary Data 9). To compare the Hi-C results with the CRISPR-based methods, 21 viruses were predicted by BLASTn-short to link with spacers in eight bins (Supplementary Data 10).

As for hybrid assembly using both Nanopore and Illumina reads, 2593 viral contigs were identified and 989 of them were deconvoluted in Hi-C data to have virus–host interactions with 144 host bins (Supplementary Data 11). To compare the Hi-C results with the CRISPR-based methods, 28 viruses were predicted by BLASTn-short to link with spacers in 10 bins (Supplementary Data 12).

Results show that CRISPR-based results have very high accuracy. For Illumina data, of the 21 virus–host connections predicted using CRISPR spacers, 11 are simultaneously found in Hi-C data and 10 are not detected in Hi-C data. Of 11 detected connections, only 1 is different in Hi-C data and 10 are the same (91% precision) (Supplementary Data 13). Also for the Nanopore/Illumina hybrid data, of the 28 virus–host connections predicted using CRISPR spacers, 16 are simultaneously found in Hi-C data and 12 are not detected in Hi-C data. Of 16 detected connections, only 1 is different in Hi-C data, 15 are the same (94% precision) (Supplementary Data 14).

It should be noticed that some of the predicted CRISPR-based virus–host interactions are undetected in Hi-C data. CRISPR spacers represent a collection of memories regarding past virus invasions, whereas Hi-C data provide a snapshot of ongoing virus–host interactions. Also, Hi-C crosslinking may not be 100% efficient and might miss some of the virus–host interactions.

## Discussion

By applying culture-independent approaches, the 50,037 metagenomic viral contigs detected in this study significantly expand current AS viromes in the databases (e.g., increasing 23-fold the IMG/VR data), allowing for a more comprehensive understanding of AS viromes. Though we have validated that de-novo assembly of viral genomes was reliable for our samples (Supplementary Data 4–6), we cannot rule out that mis-assembly or sequencing error could occasionally occur in viral genomes (estimated at ~3 non-identical ORFs per genome by comparing our predicted ORFs with reference genomes using CompareM) and might affect functional or metabolic predictions of viral genomes. Our results show that WWTPs contain very high viral diversity which is largely unknown (98.4–99.6% of total viral contigs (coverage percentage)). Despite the enormous viral diversity in AS system, the large majority of viral genera were detected in at least two of

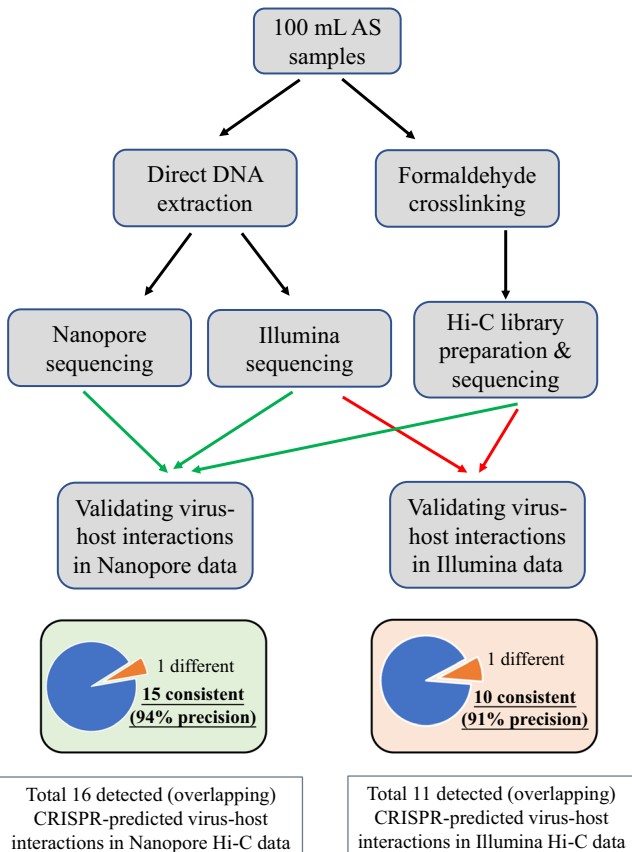

**Fig. 7 General workflow to validate the precision of CRISPR-based methods in the present study.** For Illumina data, 91% precision was observed. For Nanopore data, 94% precision was observed. Source data are provided as a Source Data file.

the samples and 53 constituted a common "core" represented in all samples (1.7–5.4% of viral contigs (coverage percentage)). Intuitively, we assumed that at a finer resolution, these viral contigs would be more site-specific. We applied MMseqs2 (easy-cluster mode)[30] to cluster 50,037 viral contigs in six WWTPs at 90% identity and 80% coverage, resulting 46043 unique viral contigs. In all, 3323 out of 46,043 (7%) unique viral contigs were shared in more than two WWTPs. The majority of viral contigs were site-specific. Because our study focused on viruses that passed through 0.2 μm filters, there may be much additional diversity (e.g., as large phages or prophages that remain integrated into the host cells). Only 614 identifiable prophages were found in our study and future work will need to focus on the recovery of phages from the cellular portion to obtain a comprehensive picture of virus diversity in AS systems. Moreover, considering the virus not being 100% recovered in virus enrichment and DNA extraction steps, and the inherent bias of Illumina genome sequencing, some underrepresented viruses might have been missed in our study. There could also be novel viral taxa or virus fragments among those sequences (~70% of the total sequences) that could not be covered by the viral contigs in this study. Nonetheless, our study suggests that it may be possible in the future to define a virome specific for AS systems within WWTPs.

A high proportion of common viral genera as well as many others could be linked to functional microorganisms in WWTPs. CRISPR-Cas spacer matches revealed that >1300 viruses may impact microorganisms in WWTPs whose functions include carbon cycling and nutrient removal. Although the exact contribution of viral predation to the mortality of functional

microorganisms remains to be investigated, viruses could potentially influence important metabolic functions through top–down control. By encoding various groups of AMGs, viruses could additionally alter carbon recycling and nutrients acquisition in their hosts. Although the CRISPR-based approach to identify hosts provides high genetic resolution, we have to bear in mind that only 50% bacterial and 90% archaeal genomes sequenced so far encode a CRISPR system[15]. Also, recombination events among viruses could evade CRISPR-Cas immune system by increasing genetic diversity[31], making it more challenging for us to track possible hosts. To supplement the CRISPR-based approach, we used Hi-C sequencing data to verify some of the virus–host links inferred by CRISPR spacer assignment and proved the precision of CRISPR-based methods. It has to be noted that Hi-C method also has its limitations, for example, the spurious links owing to shared sequence fragments between closely related bacteria or viruses. Overall, a combination of three sets of data in this study, namely targeted metagenomics, direct metagenomics, and Hi-C metagenomics, provided a holistic view of the virus–host interactions in complex ecosystems.

The high viral diversity in WWTPs may serve as a valuable source to isolate viruses specific for undesired and/or pathogenic bacteria. In our study, 402 viral genera were shown to infect *Mycobacterium*, which is a genus containing foaming bacteria abundant in WWTPs. This finding opens the possibility to isolate and apply related viruses to eliminate foaming bacteria in situ. Conventionally, disinfectants are applied to curb pathogens and control sludge bulking and foaming[32]. However, due to their non-specificity, benign microorganisms are also killed. In order to control microorganisms in a more targeted and sustainable way, phage treatment might be a means to selectively kill undesired microorganisms. For instance, two recent studies have reported the use of phages to curb foaming-associated filamentous bacteria effectively[33,34]. Moreover, from a clinical perspective, WWTP samples have already proven useful for isolating phages to kill the lethal superbug *Acinetobacter baumannii*[35]. A recent study also reported a simple and efficient technology to generate new phage genomes artificially and accelerate the process of phage treatment[36]. By isolating or synthesizing phages and applying them to remove other undesired microorganisms (e.g., foaming and bulking bacteria), stable and efficient removal of pollutants could be sustained in WWTPs.

## Methods

**Sampling, filtration, and purification.** In all, 2 L AS samples were collected, respectively, from six secondary WWTPs in Hong Kong, namely Sai Kung (SK), Sha Tin (ST), Stanley (STL), Shek Wu Hui (SWH), Tai Po (TP), and Yuen Long (YL). ST, STL, SWH, TP, and YL were sampled on 12 Dec 2018 and SK sampled on 5 Sep 2018. Sonication was applied for 15 mins to detach viruses. The samples were centrifuged at 5000 × *g* for 15 mins and viruses were collected using an iron floc-culation approach[37]. In detail, supernatants were filtered through 0.45 μm membranes (Advantec MFS, USA) and 0.22 μm Sterivex filters (Merck Millipore, USA) to remove bacteria and other contaminants. The filtrate was iron-chloride floc-culated, collected on 0.22 μm polycarbonate filters (Merck Millipore, USA), and resuspended in an oxalate solution. Polyethylene glycol was subsequently added to precipitate viruses. To remove non-encapsidated nucleic acids, Turbo DNase and RNase (Thermo Fisher Scientific, USA) were added. The samples were supplemented with ethylenediaminetetraacetic acid to inactivate nucleases.

**DNA extraction and metagenomic sequencing.** DNA was extracted using MasterPure DNA extraction kit (Epicentre, USA) with proteinase K following the manufacturer's instructions. DNA concentrations were measured using a Nano-Drop One Spectrophotometer (Thermo Fisher Scientific, USA) and DNA extracts were stored at −20°C for metagenomic sequencing. Shotgun sequencing was performed using Illumina Hiseq 4000 PE150 (for sample SK) (Novogene, China) and Illumina Novaseq 6000 PE150 (for samples ST, STL, SWH, TP, and YL) (Novogene, China). Metagenomic sequencing for SK, ST, STL, SWH, TP, and YL yielded 40.1, 32.6, 34.1, 40.6, 33.8, and 41.7 Gb raw data, respectively. Read numbers for SK, ST, STL, SWH, TP, and YL were 267 million, 217 million, 227

million, 271 million, 225 million, and 278 million, respectively. The sequencing depth (average coverage for viral contigs) for SK, ST, STL, SWH, TP, and YL was 107×, 69×, 65×, 64×, 103×, and 139×, respectively.

**Identification of metagenomic viral contigs**. The sequencing data were de-novo assembled using CLC Genomics Workbench (version 11.0.1, QIAGEN Bioinformatics, Denmark) with automatic word size and minimum scaffold length of 1 kb. Reads were mapped to contigs by CLC Genomics Workbench with length fraction 0.8 and similarity fraction 0.9 to obtain contig coverage. The assembled contigs were processed for downstream bioinformatics analysis. Two different methods[38], i.e., a modified vHMM DNA virus detection pipeline[39] and phylogeny-based pipeline[40], were used to identify metagenomic viral contigs in our study. For the vHMM DNA virus detection pipeline, genes were first predicted using Prodigal V2.6.3[41], and then aligned against Viral Protein Family Models (VPFs) database[39] and Pfam database (31.0, -cut_nc mode)[42] using hmmsearch (3.1b2)[43].

To calculate the total number of genes assigned with KEGG Orthology (KO) terms, all genes were compared with KEGG database (release 80.1)[28] using DIAMOND v0.9.25[44]. Contigs that had at least five hits with VPFs and longer than 5 kb were retained. The results were subsequently screened using three different criteria[39]. The first one is that total no. of genes for each contig aligned with KO terms was ≤20%, aligned with Pfams was ≤40%, and aligned with VPFs was ≥10%. The second one is that no. of genes for each contig aligned with VPFs was more than that for Pfams. The third one is that the number of VPFs was ≥60% of the total number of genes for each contig. Regarding the phylogeny-based pipeline, all the assembled contigs were searched against NCBI Refseq virus database V91 using DIAMOND. MEGAN6[45] was then applied to analyze taxonomy and those contigs assigned to viruses were identified as metagenomic viral contigs. Metagenomic viral contigs identified by the above-mentioned two pipelines were added together for further analyses.

**Evaluation of de-novo assembly by pooling mock community viromes**. To evaluate the de-novo assembly performance in our samples, mock community virome Illumina sequencing reads (instead of reads generated by the simulator) were downloaded from http://mirrors.iplantcollaborative.org/browse/iplant/home/shared/iVirus/DNA_Viromes_library_comparison[46]. Mock community viromes include 12 phage genomes (10 dsDNA and 2 ssDNA viruses) ranging from 5 kb to 129 kb and detailed information could be found in Supplementary Data 4. Raw reads were trimmed with Trimmomatic to remove reads quality score lower than 25 on a 4 bp sliding window with a minimum length of 150 bp[47]. Trimmed and filtered reads were pooled with each sample with a ratio ranging from 1:20 to 1:15. All pooled data were de-novo assembled using CLC Genomics Workbench (version 11.0.1, QIAGEN Bioinformatics, Denmark) and contigs were compared to reference genomes with Nucmer (4.0.0beta2)[48] and CompareM v0.0.23 (https://github.com/dparks1134/CompareM). For pooled-sample de-novo assembly, results showed that our recovered mock phage genomes had an average of 99.94% average nucleotide identity (ANI) with reference phage genomes. At the amino-acid level, our recovered phage genomes shared an average of 99.96% amino-acid identity with references (Supplementary Data 5, 6).

**Viral clustering**. Gene-sharing networks were used for viral clustering[11]. Predicted proteins from all the data sets in the present study were uploaded to CyVerse Discovery Environment and vContact2 (version 0.9.5) app was used to generate VCs by merging reference NCBI Bacterial and Archaeal Viral RefSeq V85 genomes for taxonomy assignment. Diamond was applied to calculate protein-protein similarity and protein clusters were built using MCL (The Markov Cluster Algorithm)[44]. VCs were generated by ClusterONE[49]. Those viral contigs that cannot be assigned to any VCs were classified as singletons. Results were visualized by Cytoscape (version 3.7.1).

To analyze the similarity between viral contigs in our study and other ecosystems, viral sequences and corresponding proteins from five different ecosystems, namely AS, AD, solid waste, marine, and freshwater were retrieved from IMG/VR database[10]. The number of sequences in each habitat was 2.1k, 8.5k, 5.7k, 355k, and 141k, respectively. To make it comparable with the number of sequences of this study (50k), only 50k sequences were randomly picked for data sets of marine and freshwater ecosystems for viral clustering.

**Relative abundance of viral contigs and VCs**. The mapping results were used for the calculation of the average coverage of viral contigs (bases of the aligned part of all the reads divided by the length of viral contigs). The average coverage was divided by data set size for normalization, denoted as the average normalized coverage, to determine the relative abundance against the sum of the coverages in all viral contigs. The relative abundance of VCs in AS of a WWTP was counted by adding up the relative abundance of all the viral contigs in each VC.

**Parameter optimization for CRISPR-Cas spacer analysis**. All complete viral genomes of whose hosts are bacteria and archaea ($N = 2309$) were downloaded from NCBI Refseq V91. All bacterial and archaeal assembled genomes ($N = 190,078$) from NCBI Assembly (https://www.ncbi.nlm.nih.gov/assembly) were retrieved to manually curate a CRISPR-Cas spacer database predicted by CRISPR Recognition Tool (CRT)[50] using the default parameters. Of 190,078

genomes, 90,858 (47.8%) could be predicted by CRT to contain CRISPR spacers. BLASTn-short was performed between reference viral genomes and spacers. Different coverage, identity, and e-value were tested to achieve a balance between recall percentage and precision percentage. Finally, coverage 90%, identity 97%, mismatch 1, and default e-value could result in the precision of 93.4% and recall of 22.4% (Supplementary Data 7). These parameters were applied for all the data sets in our study.

**Host prediction**. Viral contigs identified for all six WWTPs were searched against manually curated CRISPR-Cas spacer database using BLASTn-short to link viruses to their hosts with 97% identity, 90% coverage, 1 mismatch, and 1 maximum target sequence (-max_target_seqs=1). Only the best hits were retained for further analysis. These candidate prokaryotic genera were subsequently annotated with different functions in WWTPs according to their name classification in MIDAS database[21].

We have created three replicate databases randomly using RSAT-random sequence (http://rsat.sb-roscoff.fr/random-seq_form.cgi), which each contains the same number ($N = 50037$) and the same size (~614 MB) of our total viral sequences. Then we used the same criteria to select the best hit. Results showed that on average 0.70% of DNA sequences have random hits to our curated CRISPR-Cas spacer database. However, none of them could link spacers from different domains of life.

**Evaluation of the specificity of CRISPR-matching spacer hits**. To get a more general answer, we first checked reference prokaryotic viruses from NCBI Refseq database V91. Results showed that all 80,499/80,499 (100%) of the CRISPR-matching spacer hits were unique for only one virus. We also inspected our CRISPR-matching spacer hits from six WWTPs and found that 5394/5394 (100%) of spacers were unique for only one virus.

**Gene analysis**. All predicted proteins of viral contigs were compared with egg-NOG database 5.0 using eggNOG-mapper1.0.3[51]. Results were further categorized according to COG function class. ORFs affiliated to Carbohydrate-Active enZYmes Database (CAZy) were further clustered to remove redundancy and generate unique viral metabolic ORFs with CD-HIT[52]. *CobS* and *CysC* genes were further confirmed by BLASTp to NCBI nr database.

**Hi-C sequencing, direct Illumina sequencing, and Nanopore sequencing**. To make further molecular validation about the CRISPR spacer matching results, additional sampling was done in December 2020 at ST WWTP. In all, 100 mL AS sample was first centrifuged at $5000 \times g$ for 15 mins to remove supernatant. The pellets were split into two parts.

One part of pellets were resuspended with 10 mL 1% formaldehyde to crosslink virus fragments with host fragments and went through 20-min incubation. Formaldehyde-crosslinking was further quenched by glycine. After spinning down pellets, crosslinked AS sample was ground into a fine powder in a liquid nitrogen-chilled mortar. Hi-C sample was then sent to Phase Genomics (USA) for Hi-C library preparation, DNA sequencing (30 Gb), and proprietary in silico virus–host interaction reconstruction.

The other part was directly extracted for DNA using ZymoBIOMICS DNA Miniprep Kit (Zymo Research, USA). DNA concentrations were measured using a NanoDrop One Spectrophotometer (Thermo Fisher Scientific, USA) and DNA was stored at −20 °C for Illumina and Nanopore sequencing. Illumina sequencing was performed using Illumina Novaseq 6000 PE150 (Novogene, China) for 60 Gb sequencing data. The same DNA also went through in-house Nanopore sequencing, which yields 10 Gb sequencing data (Fig. 7).

Two strategies have been applied for downstream bioinformatics analysis. For the first strategy, 60 Gb Illumina sequencing data were de-novo assembled using CLC Genomics Workbench (version 11.0.1, QIAGEN Bioinformatics, Denmark) with automatic word size and minimum scaffold length of 1 kb. As for the second strategy, 10 Gb Illumina and 10 Gb Nanopore reads were hybrid-assembled using OPERA-MS (v0.8.3)[53] with pilon polishing. Then, the above-mentioned two contigs were each deconvoluted and clustered with 30 Gb Hi-C reads to achieve host genome bins and reconstruct subsequent virus–host interactions.

**Reporting summary**. Further information on research design is available in the Nature Research Reporting Summary linked to this article.

## Data availability

All raw sequencing data from the six AS viromes generated in this study have been deposited in the NCBI Sequence Read Archive (SRA) database under BioProject ID: PRJNA639411. Hi-C raw sequencing data generated in this study have been deposited in the NCBI SRA database under BioProject ID: PRJNA745436. All complete viral genomes of whose hosts are bacteria and archaea ($N = 2309$) used in this study are available in the NCBI Refseq database V91 (https://ftp.ncbi.nlm.nih.gov/refseq/release/release-catalog/archive/RefSeq-release91.catalog.gz). All bacterial and archaeal assembled genomes ($N = 190,078$) used in this study are available in the NCBI Assembly database (https://www.ncbi.nlm.nih.gov/assembly). Viral contigs from other ecosystems used in this study

are available in the IMG/VR database (https://genome.jgi.doe.gov/portal/IMG_VR/IMG_VR.home.html). Mock community virome Illumina sequencing reads used in this study are available in http://mirrors.iplantcollaborative.org/browse/iplant/home/shared/iVirus/DNA_Viromes_library_comparison. Source data are provided with this paper.

## Code availability

All scripts used in the identification of metagenomic viral contigs were deposited in https://github.com/yqchen17/Manuscript (https://doi.org/10.5281/zenodo.5180707)[54].

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

## Acknowledgements

This work was supported by Hong Kong TRS (T21-705/20-N). Mr. Y.C. would like to thank The University of Hong Kong and the Research Grants Council of Hong Kong for the Hong Kong PhD Fellowship. Dr. Y.W. wishes to thank The University of Hong Kong for the postdoc fellowship. We appreciate technical support from Ms. Vicky Fung. The computations were performed using research computing facilities offered by Information Technology Services, The University of Hong Kong. Finally, we would like to thank Ivan Liachko and Gherman Uritskiy for the technical assistance in building Hi-C library and in performing Hi-C data analysis.

## Author contributions

Y.C. and T.Z did the experimental design. Y.C. conducted the laboratory works, performed bioinformatic analysis, and prepared the manuscript. Y.W. contributed to experimental design and bioinformatic analysis. T.Z supervised the whole research and revised the manuscript. D.P.E and M.F.P contributed to the data interpretation and manuscript preparation. All authors read and approved the final paper.

## Competing interests

The authors declare no competing interests.
