## [Peer Review File · Nature Communications]

REVIEWER COMMENTS

Reviewer #1 (Remarks to the Author):

I was very pleased to have the opportunity to review this manuscript and to read of this description of the viral diversity in 6 Activated Sludge plants in Hong Kong. The editors will hopefully be aware that these seemingly prosaic environments have been critical test beds in microbial ecology and this is a paper very much in that tradition.

The paper contributes to the growing realisation that viruses are much much more important than microbial ecologists have perhaps hitherto accepted. The ongoing revolution in genomics is part of the story: the discovery of prophage and CRISPR sequences in many genomes are a clear indication that the viruses are impinging on the ecology of a wide variety of organisms.

This paper, and other papers of this nature are starting to complement this picture by assembling sequences of viral DNA into plausible genomes. There is inevitably a degree of pure observation work of this kind. But, to the authors credit, this manuscript goes further by linking the viruses to the CRISPR sequence in known genomes. This is not a perfect approach, to linking predator and prey, but it is certainly a start and a step in the right direction. However, I was wondering if the authors had considered the possibility that the sequences they are detecting are conserved in viruses and are not necessarily unique to the viruses they have sequenced?

There are also naturally some limitations, some of which the authors are aware of and highlight in the manuscript: the use of 0.2 micrometre filters and DNA sequencing natural confines their observations to small DNA viruses.

However, the authors are making a common error in metagenomics of equating abundance with proportional abundance. Whilst this is not an egregious error in the context of this manuscript, it is an insidious one for the field as a lack of quantitative tools can make progress difficult.

A case in point is that in this manuscript the authors have perhaps overlooked the fact that a significant fraction of the viruses in the mixed liquor will have come in with the wastewater and thus may be of little or no ecological significance in the reactor. This might account for the presence of viruses thought to prey on methanogens or sulphate reducing bacteria.

The manuscript is immaculately prepared and the methods used are sound.

I have the following detailed comments.

Line 81 “indicating adequate recovery” The flattening of the line is impressive. Do you think this means you have found all the viruses or just the most abundant ones?

Line 112 “from viral abundance data” There is no viral abundance data in this paper. Only proportional abundance. Both data formats have their uses, but one should not confuse them.

Line 116 “high abundance” There is no viral abundance data in this paper.

Line 131 “Overall, these results suggest that the virome across WWTPs consists of many shared genera. The lack of detection of specific genera in one AS virome may be primarily due to either detection limit or temporal dynamics. Hence, it is possible that AS common viromes may be more diverse than those observed in this study.”

I am not 100% sure that I understand this statement. But as we expect different plant to share the same bacterial taxa, surely expect them to share some viral taxa as well?

Line 194 “Intriguingly, viruses might be able to infect hosts from different domains of life, with displaying perfect matches to spacers in the CRISPR database, indicating viruses could occasionally switch hosts from bacteria to archaea.” Is there an alternative explanation? Could some viral sequences could be conserved across viruses and so they feature prominently in the CRISPR database?

Line 212 “Anaerobic processes could occur at the microenvironment within sludge flocs” This has been observed. However, about a quarter of the viruses in the mixed liquor could have come from the wastewater where methanogens (and sulphate reducers) will be abundant.

Line 224 “sulfate reducers” SRB Also in the wastewater.. especially in Hongkong where seawater is used for flushing

Line 316 “It will be important to supplement this approach with culturing-based approaches to experimentally verify virus-host links as well as to identify hosts that do not possess CRISPR-Cas defense systems.” Good point.. But not all bacteria are culturable.

Reviewer #2 (Remarks to the Author):

Chen et al present a study of prokaryotic viruses of activated sludge samples from 6 wastewater treatment plants in Hong Kong collected in 2018. They performed metagenomic sequencing and analyzed the virome using de novo assembled contigs. The authors identified 50,037 viral contigs that would be a substantial addition to existing IMG/VR database. They compared communities using viral genera and found that viral genera were largely shared across the wastewater treatment plants. The authors predict host-association by querying CRISPR-Cas spacer sequences and in turn, infer community functions indirectly or directly through auxiliary metabolic genes. There are a few major points that need to be addressed.

Major comments:

1. The AS sample from each wastewater treatment plant essentially represents large pooled environmental sources (municipality or region). The authors used de novo assembly to generate contigs for viral classification. De novo assembly is prone to creating artificial chimeric contigs that do not exist, particularly when the metagenomic dataset is from pooled environmental sources. Most of the analyses are based on the credibility of these contigs. This may also result in an over-estimation the data. Were assemblies assessed for chimeras or validated?

2. The 16S rRNA amplicon sequencing (line 109) seems abrupt. The rationale for this experiment is unclear. 16S analysis methodology is not described and 16S bacterial community data presented is fairly superficial (PCoA, Fig 1D). Authors describe that 16S rRNA sequencing data was used to determine phylum abundance in Line 159, referencing Supplementary Table 2. However, Supplementary Table 2 provided for review does not match that - “Table S2. Host prediction for shared viral genera between Sample AS and IMG/VR AS (phylum level)”. It seems that the authors use the 16S data as a validation to the CRISPR-Cas host identification approach (line 157). Therefore, it would be important to show this data to support their claim.

3. 1,668 out of 5,879 viruses mapped to multiple hosts (line 182). How did the authors handle multiple potential hosts when inferring ensemble function (Figure 4)?

4. Generally, the authors compare viral community relatedness at the genera taxonomic level (Figure 6 and 2) and conclude considerable overlap (lines 271, 119) and that viruses are extensively shared (line 287). Is this conclusion supported if done at a contig/isolate taxonomic resolution?

Minor comments:

Line 360: "Metagenomic sequencing for SK, ST, STL, SWH, TP, and YL yielded 39.7, 32.6, 34.1, 40.6, 33.8, and 41.7 Gb raw data, respectively". The sequencing read depth of each site should be included.

There are some inconsistencies in the manuscript where the authors loosely use "reads" and "contigs" interchangeably which makes it confusing. They are quite different in terms of data. For example, Line 90: "Comparison with NCBI RefSeq viral genome database showed that across the six AS systems, only between 0.4-1.6% of total viral reads could be assigned to a known viral family". Is this 0.4-1.6% of the total viral contigs? Methods do not describe reads to RefSeq.

Point-by-point response to reviewers' comments

Manuscript ID: NCOMMS-20-23333

Title: Prokaryotic viruses impact functional microorganisms in nutrient removal and carbon cycle in wastewater treatment plants

Authors: Yiqiang Chen, Yulin Wang, David Paez-Espino, Martin F. Polz, Tong Zhang

Corresponding author: Prof. Tong Zhang

Responses to comments of Reviewer 1

Reviewer #1 (Remarks to the Author):

I was very pleased to have the opportunity to review this manuscript and to read of this description of the viral diversity in 6 Activated Sludge plants in Hong Kong. The editors will hopefully be aware that these seemingly prosaic environments have been critical test beds in microbial ecology and this is a paper very much in that tradition.

The paper contributes to the growing realisation that viruses are much much more important than microbial ecologists have perhaps hitherto accepted. The ongoing revolution in genomics is part of the story: the discovery of prophage and CRISPR sequences in many genomes are a clear indication that the viruses are impinging on the ecology of a wide variety of organisms.

This paper, and other papers of this nature are starting to complement this picture by assembling sequences of viral DNA into plausible genomes. There is inevitably a degree of pure observation work of this kind. But, to the authors credit, this manuscript goes further by linking the viruses to the CRISPR sequence in known genomes. This is not a perfect approach, to linking predator and prey, but it is certainly a start and a step in the right direction. However, I was wondering if the authors had considered the possibility that the sequences they are detecting are conserved in viruses and are not necessarily unique to the viruses they have sequenced?

Responses: *Thanks for the valuable comments.*

Viral genomes are extraordinarily diverse. To get a more general answer, we first checked reference prokaryotic viruses from NCBI Refseq database V91. Results showed that all 80499/80499 (100%) of the CRISPR matching spacer hits were unique for only one virus. In this work, we also inspected our CRISPR matching spacer hits from 6 WWTPs and found that

5394/5394 (100%) of the CRISPR matched spacer hits were unique for only one virus. Therefore, we believe that it is less likely that these sequences are conserved in viruses. We have added these results in manuscript (Line 469 to 474):

“Evaluation of the specificity of CRISPR matching spacer hits

To get a more general answer, we first checked reference prokaryotic viruses from NCBI Refseq database V91. Results showed that all 80499/80499 (100%) of the CRISPR matching spacer hits were unique for only one virus. We also inspected our CRISPR matching spacer hits from 6 WWTPs and found that 5394/5394 (100%) of spacers were unique for only one virus.”

There are also naturally some limitations, some of which the authors are aware of and highlight in the manuscript: the use of 0.2 micrometre filters and DNA sequencing natural confines their observations to small DNA viruses.

Responses: *Thanks for the valuable comments. We agree that there are some limitations as we described in our manuscript.*

However, the authors are making a common error in metagenomics of equating abundance with proportional abundance. Whilst this is not an egregious error in the context of this manuscript, it is an insidious one for the field as a lack of quantitative tools can make progress difficult.

Responses: *Thanks for the valuable comments. We agree with the reviewer that absolute quantification approach should be developed in this field. In this study, we have changed all the expressions of “abundance” into “relative abundance” in our revised manuscript (Line 106, 112, 152).*

A case in point is that in this manuscript the authors have perhaps overlooked the fact that a significant fraction of the viruses in the mixed liquor will have come in with the wastewater and thus may be of little or no ecological significance in the reactor. This might account for the presence of viruses thought to prey on methanogens or sulphate reducing bacteria.

Responses: *Thanks for the valuable comments. We agree that there could be a fraction of the viruses in the mixed liquor coming in with the influent. We have checked the relative abundance of the viruses that could infect methanogens and sulfate reducing bacteria. These viruses could account for 0.27% (SK), 0.16% (ST), 0.20% (STL), 0.16% (SWH), 0.28% (TP),*

0.13% (YL) in total viral contigs (coverage percentage). These viruses might come from the wastewater.

We have added the relevant data and explanation in the manuscript. (Line 208 to 212):

“However, some viruses that display specificity for sulfate reducers (due to sea water flushing in Hong Kong) and methanogens might be detected in the AS system due to their presence in the influent. Results showed that they account for a very low percentage (0.13-0.28%) in total viral contigs (coverage percentage). These viruses might be carried from the wastewater.”

The manuscript is immaculately prepared and the methods used are sound.

Responses: *Thanks for the positive comments.*

I have the following detailed comments.

Line 81 “indicating adequate recovery” The flattening of the line is impressive. Do you think this means you have found all the viruses or just the most abundant ones?

Responses: *Thanks for the valuable comments. Considering the virus not being 100% recovered in virus enrichment and DNA extraction steps, and the inherent bias of Illumina genome sequencing, we agree that we have found just the most abundant ones and will inevitably miss some viruses in our study. In this context, we mainly investigated the impact of sequencing depth on virus recovery and the result showed that 15 Gb sequencing data would be good to recover those abundant viral contigs using the approaches in this study. Please also notice that ~70% of the total reads cannot be covered by the viral contigs in this study and may contain novel viruses or virus fragments.*

We have added these limitations in the discussion of the revised manuscript (Line 309-313):

“Moreover, considering the virus not being 100% recovered in virus enrichment and DNA extraction steps, and the inherent bias of Illumina genome sequencing, some underrepresented viruses might have been missed in our study. There could also be novel viral taxa or virus fragments among those sequences (~70% of the total sequences) that could not be covered by the viral contigs in this study.”

Line 112 “from viral abundance data” There is no viral abundance data in this paper. Only proportional abundance. Both data formats have their uses, but one should not confuse them.

Responses: *Thanks for the valuable comments. We revised all the word “abundance” to “relative abundance” to avoid confusion as suggested (Line 106, 112, 152).*

Line 116 “high abundance” There is no viral abundance data in this paper.

Responses: *Thanks for the valuable comments. We revised all the word “abundance” to “relative abundance” as suggested (Line 106, 112, 152).*

Line 131 “Overall, these results suggest that the virome across WWTPs consists of many shared genera. The lack of detection of specific genera in one AS virome may be primarily due to either detection limit or temporal dynamics. Hence, it is possible that AS common viromes may be more diverse than those observed in this study.”

I am not 100% sure that I understand this statement. But as we expect different plant to share the same bacterial taxa, surely expect them to share some viral taxa as well?

Responses: *Thanks for the valuable comments. We agree with you that there are shared viral genera among different WWTPs due to the shared prokaryotic taxa. The lack of detection of some viral genera in the AS virome of one WWTP may be primarily due to biological variation in the grab samples and/or the technical variation. Hence, if such technical and biological variations are taken into account, the virome shared among all AS may be even more diverse.*

We have revised in the manuscript accordingly (Line 127 to 130):

“The lack of detection of some viral genera in the AS virome of one WWTP may be primarily due to biological variation in the grab samples and/or the technical variation. Hence, if such technical and biological variations are taken into account, the virome shared among all AS may be even more diverse.”

Line 194 “Intriguingly, viruses might be able to infect hosts from different domains of life, with displaying perfect matches to spacers in the CRISPR database, indicating viruses could occasionally switch hosts from bacteria to archaea.” Is there an alternative explanation? Could some viral sequences could be conserved across viruses and so they feature prominently in the CRISPR database?

Responses: *Thanks for the valuable comments.*

For Q1: This was also observed in inoviruses in another paper (Nature Microbiology. 2019. 4, 1895-1906)¹. We think that further experiments may be needed to confirm this finding since we cannot completely exclude the possibility of misassembly.

For Q2: Viruses do not share a universal marker gene like the conserved ribosomal 16S rRNA genes encoded in prokaryotes². Besides, as previously mentioned, we have checked our

CRISPR matching spacer hits for 6 WWTPs and all 5394 spacer sequences were unique for only one virus. Therefore, we think it is less likely that these sequences are conserved in viruses.

We have revised in the manuscript accordingly (Line 468-473):

“Evaluation of the specificity of CRISPR matching spacer hits

To get a more general answer, we first checked reference prokaryotic viruses from NCBI Refseq database V91. Results showed that all 80499/80499 (100%) of the CRISPR matching spacer hits were unique for only one virus. We also inspected our CRISPR matching spacer hits from 6 WWTPs and found that 5394/5394 (100%) of spacers were unique for only one virus.”

Line 212 “Anaerobic processes could occur at the microenvironment within sludge flocs” This has been observed. However, about a quarter of the viruses in the mixed liquor could have come from the wastewater where methanogens (and sulphate reducers) will be abundant.

Responses: *Thanks for the valuable comments. As previously mentioned, we have checked the relative abundance of these viruses that could infect methanogens and sulfate-reducing bacteria. Results showed that they account for a very low percentage (0.13-0.28%) in total viral contigs (coverage percentage). These viruses might be carried from wastewater.*

We have added it in the manuscript (Line 208-212):

“However, some viruses that display specificity for sulfate reducers (due to sea water flushing in Hong Kong) and methanogens might be detected in the AS system due to their presence in the influent. Results showed that they account for a very low percentage (0.13-0.28%) in total viral contigs (coverage percentage). These viruses might be carried from the wastewater.”

Line 224 “sulfate reducers” SRB Also in the wastewater. especially in Hongkong where seawater is used for flushing

Responses: *Thanks for the valuable comments. We agree with the reviewer and we have added this explanation in the manuscript (Line 208 to 212):*

“However, some viruses that display specificity for sulfate reducers (due to sea water flushing in Hong Kong) and methanogens might be detected in the AS system due to their presence in the influent. Results showed that they account for a very low percentage (0.13-0.28%) in total viral contigs (coverage percentage). These viruses might be carried from the wastewater.”

Line 316 “It will be important to supplement this approach with culturing-based approaches to experimentally verify virus-host links as well as to identify hosts that do not possess CRISPR-Cas defense systems.” Good point. But not all bacteria are culturable.

***Responses:** Thanks for the valuable comments. We agree with you. Some advanced methods which are being developed, such as single-cell genome sequencing and chromosome conformation capture method, could be used to verify the virus-host relationship in the future. We have added the relevant discussion into the revised manuscript (Line 327-330):*

“It will be important to supplement this approach with culturing-based approaches, single-cell genome sequencing or chromosome conformation capture method to experimentally verify virus-host links as well as to identify hosts that do not possess CRISPR-Cas defense systems.”

Response to comments of Reviewer 2

Reviewer #2 (Remarks to the Author):

Chen et al present a study of prokaryotic viruses of activated sludge samples from 6 wastewater treatment plants in Hong Kong collected in 2018. They performed metagenomic sequencing and analyzed the virome using de novo assembled contigs. The authors identified 50,037 viral contigs that would be a substantial addition to existing IMG/VR database. They compared communities using viral genera and found that viral genera were largely shared across the wastewater treatment plants. The authors predict host-association by querying CRISPR-Cas spacer sequences and in turn, infer community functions indirectly or directly through auxiliary metabolic genes. There are a few major points that need to be addressed.

***Responses:** Thanks for the general comments.*

Major comments:

1. The AS sample from each wastewater treatment plant essentially represents large pooled environmental sources (municipality or region). The authors used de novo assembly to generate contigs for viral classification. De novo assembly is prone to creating artificial chimeric contigs that do not exist, particularly when the metagenomic dataset is from pooled environmental sources. Most of the analyses are based on the credibility of these contigs. This may also result in an over-estimation the data. Were assemblies assessed for chimeras or validated?

Responses: *Thanks for the valuable comments. We added an in-silico analysis to validate the contig assembly. Firstly, we downloaded the Illumina sequencing reads (instead of reads generated by the simulator) of 12 phage mock community genomes³, including 10 dsDNA viruses and 2 ssDNA viruses ranging from 5kb to 129 kb, and tested contig assembly performance. Secondly, mock community datasets were pooled with each activated sludge sample of the 6 WWTPs at a ratio of 1:15-1:20 to evaluate the recovery of these mock community in the complex samples.*

For mock-community-only contig assembly, results showed that the recovered mock phage genomes had an average of 99.91% average nucleotide identity (ANI) with reference phage genomes. At amino acid level, our recovered phage genomes shared an average of 99.97% amino acid identity (AAI) with references.

For pooled-sample contig assembly, results showed that the recovered mock phage genomes had an average of 99.94% average nucleotide identity (ANI) with reference phage genomes. At amino acid level, the recovered phage genomes shared an average of 99.96% amino acid identity (AAI) with references.

Considering the inherent sequencing error caused by Illumina sequencing (~99.9% accuracy), the de novo assembly for phage genomes in the complex dataset is reliable and the probability of generating chimeric contigs in our studies is low.

Nevertheless, we agree that we might still mis-annotate a small part of genes (~3 non-identical ORFs per genome) in the phage genomes due to misassemblies or sequencing errors.

We have added this limitation in the discussion (Line 289-294) of the revised manuscript and added relevant detailed analysis in the methods part (Line 405-422) and the supplementary tables (Supplementary Table 4-6):

“Though we have validated that de novo assembly of viral genomes was reliable for our samples (Supplementary Table 4-6), we cannot rule out that mis-assembly or sequencing error could occasionally occur in viral genomes (estimated at ~3 non-identical ORFs per genome by comparing our predicted ORFs with reference genomes using CompareM) and might affect functional or metabolic predictions of viral genomes.”

“Evaluation of de-novo assembly by pooling mock community viromes

To evaluate the de-novo assembly performance in our samples, mock community virome Illumina sequencing reads (instead of reads generated by the simulator) were downloaded from http://mirrors.iplantcollaborative.org/browse/iplant/home/shared/iVirus/DNA_Viromes_library_comparison. Mock community viromes include 12 phage genomes (10 dsDNA and 2 ssDNA viruses) ranging

from 5 kb to 129 kb and detailed information could be found in Supplementary Table 4. Raw reads were trimmed with Trimmomatic to remove reads quality score lower than 25 on a 4 bp sliding window with a minimum length of 150 bp. Trimmed and filtered reads were pooled with each sample with a ratio ranging from 1:20-1:15. All pooled data were de novo assembled using CLC Genomics Workbench (version 11.0.1, QIAGEN Bioinformatics, Denmark) and contigs were compared to reference genomes with Nucmer and CompareM (<https://github.com/dparks1134/CompareM>). For pooled-sample de novo assembly, results showed that our recovered mock phage genomes had an average of 99.94% average nucleotide identity (ANI) with reference phage genomes. At amino acid level, our recovered phage genomes shared an average of 99.96% amino acid identity (AAI) with references (Supplementary Table 5-6).”

2. The 16S rRNA amplicon sequencing (line 109) seems abrupt. The rationale for this experiment is unclear. 16S analysis methodology is not described and 16S bacterial community data presented is fairly superficial (PCoA, Fig 1D). Authors describe that 16S rRNA sequencing data was used to determine phylum abundance in Line 159, referencing Supplementary Table 2. However, Supplementary Table 2 provided for review does not match that - “Table S2. Host prediction for shared viral genera between Sample AS and IMG/VR AS (phylum level)”. It seems that the authors use the 16S data as a validation to the CRISPR-Cas host identification approach (line 157). Therefore, it would be important to show this data to support their claim.

Responses: *Thanks for the valuable comments. We are sorry that 16S rRNA amplicon data caused some misunderstanding. We have deleted the parts related to 16S rRNA amplicon sequencing in the revised manuscript (Figure 1d, Line 111, Line 708-709) and supplementary table (Supplementary Table 2).*

We did not use 16S data as validation for the CRISPR-Cas host identification. Instead, we performed the validation of CRISPR-Cas host assigning approach based on the NCBI Refseq prokaryotic genomes V91 as described in the Methods part (Line 449-458):

“Parameter optimization for CRISPR-Cas spacer analysis

All complete viral genomes of whose hosts are bacteria and archaea (N=2309) were downloaded from NCBI Refseq V91. All bacteria and archaea genomes (N=190,078) from NCBI Assembly were retrieved to manually curate a CRISPR-Cas spacer database predicted by CRISPR Recognition Tool (CRT). BLASTn-short was performed between reference viral

genomes and spacers. Different coverage, identity and e-value were tested to achieve a balance between recall percentage and precision percentage. Finally, coverage 90%, identity 97%, mismatch 1 and default e-value could result in precision of 93.4% and recall of 22.4%. These parameters were applied for all the datasets in our study.”

3. 1,668 out of 5,879 viruses mapped to multiple hosts (line 182). How did the authors handle multiple potential hosts when inferring ensemble function (Figure 4)?

Responses: *Thanks for the valuable comments. For those broad-host viruses, we assumed that they could indirectly impact multiple functions in WWTPs. For example, viral contig STL1812_19787 could infect both Candidatus Accumulibacter (phosphate-accumulating organism) and Desulfosoma (sulfate-reducing organism). By infecting these two hosts, STL1812_19787 could indirectly impact both phosphate removal process and sulfate reducing process. In this case, we add one to the count of both Candidatus Accumulibacter and Desulfosoma in the Figure 4.*

We have added the relevant description into the revised manuscript. Please refer to Line 188-194:

“For those broad-host viruses, we assumed that they could indirectly impact multiple functions in WWTPs. For example, viral contig STL1812_19787 could infect both Candidatus Accumulibacter (phosphate-accumulating organism) and Desulfosoma (sulfate-reducing organism). By infecting these two hosts, STL1812_19787 could indirectly impact both phosphate removal process and sulfate reducing process. In this case, we add one to the count of both Candidatus Accumulibacter and Desulfosoma in Figure 4.”

4. Generally, the authors compare viral community relatedness at the genera taxonomic level (Figure 6 and 2) and conclude considerable overlap (lines 271, 119) and that viruses are extensively shared (line 287). Is this conclusion supported if done at a contig/isolate taxonomic resolution?

Responses: *Thanks for the valuable comments. Intuitively, we assumed that at a finer resolution, these viral contigs would be more site-specific. We applied MMseqs2 (easy-cluster mode)⁴ to cluster 50037 viral contigs in 6 WWTPs at 90% identity and 80% coverage, resulting 46043 unique viral contigs. 3323 out of 46043 (7%) unique viral contigs were shared in more than two WWTPs. The majority of viral contigs were site-specific.*

We have added this into the revised manuscript. Please refer to Line 299-303:

“Intuitively, we assumed that at a finer resolution, these viral contigs would be more site-specific. We applied MMseqs2 (easy-cluster mode) to cluster 50037 viral contigs in 6 WWTPs at 90% identity and 80% coverage, resulting 46043 unique viral contigs. 3323 out of 46043 (7%) unique viral contigs were shared in more than two WWTPs. The majority of viral contigs were site-specific.”

Minor comments: Line 360: “Metagenomic sequencing for SK, ST, STL, SWH, TP, and YL yielded 39.7, 32.6, 34.1, 40.6, 33.8, and 41.7 Gb raw data, respectively”. The sequencing read depth of each site should be included.

***Responses:** Thanks for the valuable comments. We used PE150 for Illumina sequencing. Read numbers for SK, ST, STL, SWH, TP, and YL were 267 million, 217 million, 227 million, 271 million, 225 million and 278 million, respectively. The sequencing depth (average coverage for viral contigs) for SK, ST, STL, SWH, TP, and YL was 107×, 69×, 65×, 64×, 103×, 139×, respectively.*

We have added the read number and sequencing depth (average coverage for viral contigs) in Line 373-377:

“Read numbers for SK, ST, STL, SWH, TP, and YL were 267 million, 217 million, 227 million, 271 million, 225 million and 278 million, respectively. The sequencing depth (average coverage for viral contigs) for SK, ST, STL, SWH, TP, and YL was 107×, 69×, 65×, 64×, 103× and 139×, respectively.”

There are some inconsistencies in the manuscript where the authors loosely use “reads” and “contigs” interchangeably which makes it confusing. They are quite different in terms of data. For example, Line 90: “Comparison with NCBI RefSeq viral genome database showed that across the six AS systems, only between 0.4-1.6% of total viral reads could be assigned to a known viral family”. Is this 0.4-1.6% of the total viral contigs? Methods do not describe reads to RefSeq.

***Responses:** Thanks for the valuable comments. Read percentage considers the coverage of each viral contig, while contig percentage only considers the contig number. To avoid misunderstanding, we have changed all the expression of “total viral reads” into “total viral contigs” and we have distinguished “type percentage” with “coverage percentage” (Line 59-60, 91, 96-97, 117, 143-144, 167, 187-188, 211, 295-296, 298). For example, if we discover 10 actinophages that each has coverage of 4 and 2 coliphages that each has*

coverage of 30, then, coliphages account for $2/12=17\%$ of total viral contigs (type percentage), but account for $2*30/100=60\%$ of total viral contigs (coverage percentage).

We performed taxonomy assignment in viral clustering step (Line 425-429) and viral coverage was obtained from CLC mapping (Line 382-384).

“Predicted proteins from all the datasets in the present study were uploaded to CyVerse Discovery Environment and vContact2 (version 0.9.5) app was used to generate viral clusters (VCs) by merging reference NCBI Bacterial and Archaeal Viral RefSeq V85 genomes for taxonomy assignment.”

“Reads were then mapped to contigs by CLC Genomics Workbench with length fraction 0.8 and similarity fraction 0.9 to obtain contig coverage.”

Reference:

1. Roux S, *et al.* Cryptic inoviruses revealed as pervasive in bacteria and archaea across Earth's biomes. *Nat Microbiol* **4**, 1895-1906 (2019).
2. Roux S, *et al.* Minimum Information about an Uncultivated Virus Genome (MIUViG). *Nat Biotechnol* **37**, 29-37 (2019).
3. Roux S, *et al.* Towards quantitative viromics for both double-stranded and single-stranded DNA viruses. *PeerJ* **4**, e2777 (2016).
4. Steinegger M, Söding J. MMseqs2 enables sensitive protein sequence searching for the analysis of massive data sets. *Nature Biotechnology* **35**, 1026-1028 (2017).

REVIEWER COMMENTS

Reviewer #1 (Remarks to the Author):

I was very grateful for the opportunity to reevaluate this manuscript in the light of the comments that I and other reviewers have made. I am delighted to see that the authors have taken these comments to heart and addressed my queries in a cogent and comprehensive manner. I am sorry to have taken a little while to communicate this good opinion.

Reviewer #2 (Remarks to the Author):

Authors have made substantial clarifications that improved the manuscript. Regarding the major comments from the first review, most have been addressed sufficiently except for one remaining major concern:

1. Regarding host assignment issues (previous major comment #3), is the basis for predicting connections between the virus to metabolic functions ("Viruses may impact key functional microorganisms in WWTPS section, beginning from new line 180). Authors provided clarification to the approach that it is handled in a fairly simplistic way (counting both each – new lines 188-194). Reviewer notes that while this is not "inaccurate"; however, this approach is not particularly innovative either. No molecular validation was provided to substantiate the multi-host interactions. This could potentially lead to over-estimation (as in line 198, 204, 214). Hence, the connections noted by the authors could be considered speculative (Figure 4). In this section, the authors used words such as "may", "might", "suggest" throughout which is in line with the descriptive nature. Whereas reviewer/readers are expecting more substantial insights into these interactions. Overall, this is the weakest section of the manuscript in terms of impact.

Comments that authors have resolved:

The in silico validation using mock communities (in response to previous major comment #1), was assessed thoroughly. Reviewer finds the data convincing. Major comment #1 is resolved.

(Previous major comment #2) Regarding 16S data. Authors removed that section in this revision, which reviewer agrees is appropriate. Major comment #2 is resolved.

(Previous major comment #4) Regarding site-specific contigs. Authors clarified the methodology. Major comment #4 is resolved.

Reviewer #3 (Remarks to the Author):

I was contacted to review only the quality of the CRISPR based host prediction.

I think that the approach seems correct, but there is no statistical assessment of spacer matches to viruses. In other words, how many hits would be expected randomly, using the same approach, taking into consideration the size and number of all sequences? How does this number compare to the actual number of predicted pairs? I think this information is important to assess the results.

One way of doing that could be the e-value threshold in the blastn searches. The e-value threshold multiplied by the number of queries could give an estimation of the virus-host pairs expected only by chance.

As you use different requirements apart from an e-value, you could be interested in a different approach. A good way is to create databases with the same number and sizes of your sequences but created randomly. And then use the same criteria for the selection of hits.

This expected number of random pairs must be taking into account for certain conclusions, i.e. is 135 viruses infecting different domains of life far from expected randomly? Of course, it is possible to make two sets (or more) of requirements (one set more restrictive than the other) that can help to support or discard the predictions made. Also, it is possible to calculate, for instance, the expected number of random hits of the subset of archaeal spacers to the subset of viruses predicted to infect bacteria.

Another issue that I would recommend is to turn on the blast filter for redundant sequences. If this filter was used, which is probably true if you only obtained unique matches to spacers, describe it in the methods.

When considering the effect of viruses in hosts with a particular metabolism, it could also be useful to compare the number of spacers in hosts with the given metabolism. That could allow the evaluation of metabolic pathways that are usually not associated with many CRISPR spacers. Or the opposite, find that some metabolisms are associated with a disproportionately large number of unique spacers.

Lines 462-463: Can more details about this curated database be explained? Did the authors use another not cited database or construct a new one? What were the general criteria for curating CRISPR-Cas spacers? What microorganisms were included in the database? Does the database represent all prokaryotic diversity or just WWTP genomes? How many organisms were analyzed for the database? How many of those organisms have CRISPR spacers?

Point-by-point response to the reviewers' comments

Manuscript ID: NCOMMS-20-23333B

Title: Prokaryotic viruses impact functional microorganisms in nutrient removal and carbon cycle in wastewater treatment plants

Authors: Yiqiang Chen, Yulin Wang, David Paez-Espino, Martin F. Polz, Tong Zhang

Corresponding author: Prof. Tong Zhang

We highly appreciate the valuable comments from reviewers. We have carefully revised our manuscript according to these comments. The revisions have been highlighted with yellow color. The point-by-point responses are presented as follows.

Response to comments of Reviewer 1

Reviewer #1 (Remarks to the Author):

I was very grateful for the opportunity to reevaluate this manuscript in the light of the comments that I and other reviewers have made. I am delighted to see that the authors have taken these comments to heart and addressed my queries in a cogent and comprehensive manner. I am sorry to have taken a little while to communicate this good opinion.

Responses: *Thanks for the positive comments.*

Response to comments of Reviewer 2

Reviewer #2 (Remarks to the Author):

Authors have made substantial clarifications that improved the manuscript. Regarding the major comments from the first review, most have been addressed sufficiently except for one remaining major concern:

1. Regarding host assignment issues (previous major comment #3), is the basis for predicting connections between the virus to metabolic functions (“Viruses may impact key functional microorganisms in WWTPS section, beginning from new line 180). Authors provided clarification to the approach that it is handled in a fairly simplistic way (counting both each –

new lines 188-194). Reviewer notes that while this is not “inaccurate”; however, this approach is not particularly innovative either. No molecular validation was provided to substantiate the multi-host interactions. This could potentially lead to over-estimation (as in line 198, 204, 214). Hence, the connections noted by the authors could be considered speculative (Figure 4). In this section, the authors used words such as “may”, “might”, “suggest” throughout which is in line with the descriptive nature. Whereas reviewer/readers are expecting more substantial insights into these interactions. Overall, this is the weakest section of the manuscript in terms of impact.

Responses: *Thanks for the valuable comments.*

We acknowledge that over-estimation of virus-host interactions might happen in our study and possibility of the over-estimation could be further assessed in another research, using those emerging methods to verify the virus-host relationships, including single-cell genome sequencing and chromosome conformation capture method like Hi-C. We hope that the reviewer recognizes that this work is beyond the current manuscript as it focuses on documenting the extensive diversity of the viruses in WWTPs.

Nonetheless, we want to provide a putative virus-host connections catalog for the potential application of phage treatment in WWTPs. Without this catalog, it would still be hard to deconvolute Hi-C data.

We have added the limitations of our work and some future possible developments in the manuscript (Line 334 to 337):

It will be important to supplement this approach with culturing-based approaches, single-cell genome sequencing or chromosome conformation capture method to experimentally verify virus-host links as well as to identify hosts that do not possess CRISPR-Cas defense systems.

Comments that authors have resolved:

The *in silico* validation using mock communities (in response to previous major comment #1), was assessed thoroughly. Reviewer finds the data convincing. Major comment #1 is resolved.

Responses: *Thanks for the positive comments.*

(Previous major comment #2) Regarding 16S data. Authors removed that section in this revision, which reviewer agrees is appropriate. Major comment #2 is resolved.

Responses: *Thanks for the positive comments.*

(Previous major comment #4) Regarding site-specific contigs. Authors clarified the methodology. Major comment #4 is resolved.

Responses: *Thanks for the positive comments.*

Response to comments of Reviewer 3

Reviewer #3 (Remarks to the Author):

I was contacted to review only the quality of the CRISPR based host prediction.

I think that the approach seems correct, but there is no statistical assessment of spacer matches to viruses. In other words, how many hits would be expected randomly, using the same approach, taking into consideration the size and number of all sequences? How does this number compare to the actual number of predicted pairs? I think this information is important to assess the results.

One way of doing that could be the e-value threshold in the blastn searches. The e-value threshold multiplied by the number of queries could give an estimation of the virus-host pairs expected only by chance.

Responses: *Thanks for the valuable comments.*

As for the e-value, actually we used the blastn default e-value 10. From NCBI website (https://blast.ncbi.nlm.nih.gov/Blast.cgi?CMD=Web&PAGE_TYPE=BlastDocs&DOC_TYPE=FAQ), it is said that virtually identical short alignments have relatively high E values, which is exactly the case in our study since we use blastn-short function.

We set the coverage to be stable as 90%, identity as 97%, mismatch as 1 and analyzed different e-values (from 1e-10 to 10). Then we tested the precision and recall for the NCBI Refseq Prokaryotic virus dataset (V91). Our results showed that stricter e-value parameter will not largely improve the precision rate (from 93.4% to 98.6%), but it will significantly decrease the virus recall (from 22.4% to 13.5%) in the test dataset. For example, in the real ST1812 sample, if we lower the e-value to 1e-10, we could only get 9 matched host genera compared with 414 host genera found at default e-value. Therefore, considering the trade-off between precision and recall, we mainly controlled other parameters like identity, coverage and mismatch.

We have added the precision and recall results regarding e-value, precision, coverage parameters for the NCBI Refseq Prokaryotic virus dataset (V91) in Supplementary Table 7.

As you use different requirements apart from an e-value, you could be interested in a different approach. A good way is to create databases with the same number and sizes of your sequences but created randomly. And then use the same criteria for the selection of hits.

This expected number of random pairs must be taking into account for certain conclusions, i.e. is 135 viruses infecting different domains of life far from expected randomly? Of course, it is possible to make two sets (or more) of requirements (one set more restrictive than the other) that can help to support or discard the predictions made. Also, it is possible to calculate, for instance, the expected number of random hits of the subset of archaeal spacers to the subset of viruses predicted to infect bacteria.

Responses: *Thanks for the valuable comments.*

Following your suggestion, we have created a database randomly using RSAT-random sequence (http://rsat.sb-roscoff.fr/random-seq_form.cgi) which contain the same number (N=50037) and the same size (~614MB) of our total viral sequences. Then we used the same criteria to select best hit. Results showed that 340/50037 (0.68%) random DNA sequences have hits to our curated CRISPR-Cas spacer database. However, none of them could link spacers from different domains of life.

Therefore, our conclusion about 135 viruses infecting different domains of life is unlikely to happen simply by chance.

We have added this part in the methods (Line 476-481):

We have created a database randomly using RSAT-random sequence (http://rsat.sb-roscoff.fr/random-seq_form.cgi) which contain the same number (N=50037) and the same size (~614MB) of our total viral sequences. Then we used the same criteria to select best hit. Results showed that 340/50037 (0.68%) random DNA sequences have hits to our curated CRISPR-Cas spacer database. However, none of them could link spacers from different domains of life.

Another issue that I would recommend is to turn on the blast filter for redundant sequences. If this filter was used, which is probably true if you only obtained unique matches to spacers, describe it in the methods.

Responses: *Thanks for the valuable comments.*

Yes, we turned on the blast filter. Following your suggestion, we have described it in the methods (Line 470-472):

Viral contigs identified for all six WWTPs were searched against manually curated CRISPR-Cas spacer database using BLASTn-short to link viruses to their hosts with 97% identity, 90% coverage, 1 mismatch and 1 maximum target sequence.

When considering the effect of viruses in hosts with a particular metabolism, it could also be useful to compare the number of spacers in hosts with the given metabolism. That could allow the evaluation of metabolic pathways that are usually not associated with many CRISPR spacers. Or the opposite, find that some metabolisms are associated with a disproportionately large number of unique spacers.

Responses: *Thanks for the valuable comments.*

We have checked the number of spacers in all Midas genera (functional microorganisms in WWTPs) in our study. Results showed that these Midas genera contain on average 44 spacers in their genomes. Notably, for those genera that have more than 10 reference genomes in our database, Methanosarcina, Sorangium, Desulfotomaculum and Methanoculleus have more than 100 spacers in their genomes, indicating their strong potential virus-host interactions (Supplementary Table 8).

We have added this part in the revised manuscript (Line 226-231) and Supplementary Table 8:

We have also checked the number of spacers in all Midas genera (functional microorganisms in WWTPs) in our study. Results showed that these Midas genera contain on average 44 spacers in their genome. Notably, for those genera that have more than 10 reference genomes in our curated database, Methanosarcina, Sorangium, Desulfotomaculum and Methanoculleus have more than 100 spacers in their genomes, indicating their strong potential virus-host interactions (Supplementary Table 8).

Lines 462-463: Can more details about this curated database be explained? Did the authors use another not cited database or construct a new one? What were the general criteria for curating CRISPR-Cas spacers? What microorganisms were included in the database? Does the database represent all prokaryotic diversity or just WWTP genomes? How many organisms were analyzed for the database? How many of those organisms have CRISPR spacers?

Responses: *Thanks for your question which we need to further explain.*

Did the authors use another not cited database or construct a new one?

We have constructed a new database on our own.

What were the general criteria for curating CRISPR-Cas spacers?

We used the default parameter of CRT¹ to predict CRISPR-Cas spacers for all 190078 prokaryotic genomes. For CRISPRs with Cas genes, CRT could achieve 99% precision and 99% recall.

What microorganisms were included in the database? Does the database represent all prokaryotic diversity or just WWTP genomes? How many organisms were analyzed for the database?

We downloaded all assembled bacterial and archaeal genomes (N=190078) from NCBI assembly website (<https://www.ncbi.nlm.nih.gov/assembly>), so we think this database could represent all prokaryotic diversity.

How many of those organisms have CRISPR spacers?

Of 190078 genomes, 90858 (47.8%) could be predicted by CRT to contain CRISPR spacers.

Can more details about this curated database be explained?

We have added these details in the revised manuscript to give clear information (Line 458 to 467):

All bacterial and archaeal assembled genomes (N=190,078) from NCBI Assembly (<https://www.ncbi.nlm.nih.gov/assembly>) were retrieved to manually curate a CRISPR-Cas spacer database predicted by CRISPR Recognition Tool (CRT). Of 190078 genomes, 90858 (47.8%) could be predicted by CRT to contain CRISPR spacers. BLASTn-short was performed between reference viral genomes and spacers. Different coverage, identity and e-value were tested to achieve a balance between recall percentage and precision percentage. Finally, coverage 90%, identity 97%, mismatch 1 and default e-value could result in precision of 93.4% and recall of 22.4% (Supplementary Table 7). These parameters were applied for all the datasets in our study.

Reference:

1. Bland C, *et al.* CRISPR recognition tool (CRT): a tool for automatic detection of clustered regularly interspaced palindromic repeats. *BMC Bioinformatics* **8**, 209 (2007).

Reviewers' comments:

Reviewer #2 (Remarks to the Author):

In the prior round of comments to the authors, Reviewer's only concern was the descriptive nature of the virus-host interactions. As explained, there are concerns of over-estimation in these data from wastewater treatment plant samples. No molecular validation was provided to substantiate the virus-host interactions. In this revision, Authors have decided to address this concern by stating that this is a limitation of their study in the Discussion (lines 334-337). No further new analyses/experiments were done to address this concern.

In the Reviewer Assessment, Reviewer is asked to evaluate: "What are the major claims of the paper? Are they novel and will they be of interest to others in the community and the wider field?" The reporting of 50,037 viral contigs from environmental sampling is interesting, but not substantive by itself. The potentially novel insights from virus-host interactions would have clearly elevated this manuscript to be "of interest to others in the community and the wider field". Reviewer also considered whether the methodology could justify. The overall approach is standard, but not particularly novel - to be clear, this is not a major critique of the study.

Based on these, Reviewer's opinion is that the current form of the manuscript may be of limited interest "to others in the community and the wider field".

Reviewer #3 (Remarks to the Author):

The authors have made good progress towards heeding my suggestions, however certain small issues remain.

Lines 470-472. Contrary to what the response to the review says, there is no mention of having turned on the blast filter for redundant sequences.

Lines 266-231. My comment was about how the biases due to an uneven distribution of CRISPR in prokaryotic groups can affect the results. The abundance of CRISPR spacers doesn't necessarily correlate with how often prokaryotes are attacked by viruses or the number of different viruses

attacking the prokaryote. Therefore, I don't think that the phrase "strong potential virus-host interactions" is as adequate, instead there is a higher probability to detect virus-host interactions. If we applied this to particular metabolisms, for some it will be easier to find corresponding viruses. Yet, it is possible to find many viruses targeting a given metabolism and still have a lot of un-affected strains. The opposite can also be true.

Line 461: This is the place to include, about CRT, something as "using the default parameters".

Line 479: Compare the 0.68% of recall in random sequences to your recall in the datasets for the same real sequences. Is real recall 11-22.6% (line 143)? It could also be stated in line 143 that this implies that the number of erroneous associations can be between 6% ($100 \times 0.68 / 11$) and 3% ($100 \times 0.68 / 22.6$). Also, the process of testing against random sequences is better if more than one replicate is used.

-Lines 480-481." However, none of them could link spacers from different domains of life."

The fact that, with random viral sequences, no domains of life were linked is not representative for the exposed results. As with random sequences the number of associations is expectedly decreased, it is to be expected that the random chance of associating spacers from different domains of life is greatly reduced. For instance, if recall is 22,4% of viruses compared to 0,68%, there is 32 times more opportunities to have an incorrect association to an already paired viral genome in your actual data.

A much adequate approach for that purpose would be to take only the recalled viral genomes, transform them into random sequences, and see recall to the set of spacers from the other domain. It is a good practice to repeat this process a few times, not just one. And then compare to the number of cross-domain pairings obtained. However, this approach has a few pitfalls (viruses that were recalled for two domains and a percentage of the recalls may be incorrect).

Another possibility is to take the estimated percentage of erroneous host identifications (see commentary about line 479). For example, let's assume an estimate of 3% errors and every virus predicted to infect 2 domains that has two identifications (one for each domain). So the possibility that at least one of these identifications is false equals $P(f) = 1 - (1 - 0.03)^2 = 0,059 = 5,9\%$; Therefore, of 135 viruses infecting 2 domains of live, about 8.0 (5.9%) would be expected to have happened just by chance.

Point-by-point response to the reviewers' comments

Manuscript ID: NCOMMS-20-23333C-Z

Title: Prokaryotic viruses impact functional microorganisms in nutrient removal and carbon cycle in wastewater treatment plants

Authors: Yiqiang Chen, Yulin Wang, David Paez-Espino, Martin F. Polz, Tong Zhang

Corresponding author: Prof. Tong Zhang

We highly appreciate the valuable comments from reviewers. This time we have substantially revised our manuscript according to these comments, especially with an additional cutting-edge high throughput chromosome conformation capture (Hi-C) sequencing to validate some of the potential virus-host connections and proved the precision of CRISPR-based methods. The revisions have been highlighted with yellow color. The point-by-point responses are presented as follows.

Response to comments of Reviewer 2

Reviewer #2 (Remarks to the Author):

In the prior round of comments to the authors, Reviewer's only concern was the descriptive nature of the virus-host interactions. As explained, there are concerns of over-estimation in these data from wastewater treatment plant samples. No molecular validation was provided to substantiate the virus-host interactions. In this revision, Authors have decided to address this concern by stating that this is a limitation of their study in the Discussion (lines 334-337). No further new analyses/experiments were done to address this concern.

In the Reviewer Assessment, Reviewer is asked to evaluate: "What are the major claims of the paper? Are they novel and will they be of interest to others in the community and the wider field?" The reporting of 50,037 viral contigs from environmental sampling is interesting, but not substantive by itself. The potentially novel insights from virus-host interactions would have clearly elevated this manuscript to be "of interest to others in the community and the wider field". Reviewer also considered whether the methodology could justify. The overall approach is standard, but not particularly novel - to be clear, this is not a major critique of the study.

Based on these, Reviewer's opinion is that the current form of the manuscript may be of limited interest "to others in the community and the wider field".

Responses: *Thanks for the valuable comments.*

To address the only concern raised by Reviewer #2 regarding the molecular validation of the CRISPR-based host assignment for viruses, we did an additional sampling in 2020-12 at Shatin WWTP (one of the sites mentioned in the paper) and used the novel high throughput chromosome conformation capture (Hi-C) method to validate virus-host connections in the sample. To validate the virus-host connections predicted by our CRISPR-based methods, we also used Illumina and Nanopore sequencing to obtain viral contigs and host genome bins in this sample (Figure 7).

As for Illumina metagenomic sequencing, 4578 viral contigs were identified and 1695 of them were deconvoluted in Hi-C data to have virus-host interactions with 197 host bins (Supplementary Table 9). To compare the Hi-C results with the CRISPR-based methods, 21 viruses were predicted by BLASTn-short to link with spacers in 8 bins (Supplementary Table 10).

As for hybrid assembly using both Nanopore and Illumina reads, 2593 viral contigs were identified and 989 of them were deconvoluted in Hi-C data to have virus-host interactions with 144 host bins (Supplementary Table 11). To compare the Hi-C results with the CRISPR-based methods, 28 viruses were predicted by BLASTn-short to link with spacers in 10 bins (Supplementary Table 12).

Our results show CRISPR-based results have a very high accuracy. For Illumina data, of the 21 virus-host connections predicted using CRISPR spacers, 11 are simultaneously found in Hi-C data and 10 are not detected in Hi-C data. Of 11 detected connections, only 1 is different in Hi-C data and 10 are the same (91% precision) (Supplementary Table 13). Also for the Nanopore/Illumina hybrid data, of the 28 virus-host connections predicted using CRISPR spacers, 16 are simultaneously found in Hi-C data and 12 are not detected in Hi-C data. Of 16 detected connections, only 1 is different in Hi-C data, 15 are the same (94% precision) (Supplementary Table 14).

It should be noticed that some of the predicted CRISPR-based virus-host interactions are undetected in Hi-C data. CRISPR spacers represent a collection of memories regarding past virus invasions, while Hi-C data provide a snapshot of ongoing virus-host interactions. Also,

Hi-C crosslinking may not be 100% efficient and might miss some of the virus-host interactions.

Overall, our further new molecular validation experiments confirmed some of the virus-host connections in the WWTP and proved the precision of CRISPR-based methods.

For raw Illumina/Nanopore/Hi-C sequencing data, we have uploaded to Google Drive: https://drive.google.com/drive/folders/1gQwMjAXpycc5tnu2Tdm_tlghj2EYfVOq?usp=sharing. These data have been uploaded to NCBI Sequence Read Archive (SRA) database (BioProject ID: PRJNA745436).

We have added the relevant Hi-C sequencing parts and Figure 7 in the manuscript (Line 300-328, Line 372-378, Line 538-563, Line 567-569, Line 864-869):

Hi-C validation of virus-host interactions in AS system

High throughput chromosome conformation capture (Hi-C) method was used to validate the virus-host connections predicted by our CRISPR-based methods using an additional sample in December 2020 at ST WWTP, by referring to viral contigs and host genome bins obtained from direct sequencing using Illumina and Nanopore metagenomic sequencing (Figure 7).

As for Illumina metagenomic sequencing, 4578 viral contigs were identified and 1695 of them were deconvoluted in Hi-C data to have virus-host interactions with 197 host bins (Supplementary Table 9). To compare the Hi-C results with the CRISPR-based methods, 21 viruses were predicted by BLASTn-short to link with spacers in 8 bins (Supplementary Table 10).

As for hybrid assembly using both Nanopore and Illumina reads, 2593 viral contigs were identified and 989 of them were deconvoluted in Hi-C data to have virus-host interactions with 144 host bins (Supplementary Table 11). To compare the Hi-C results with the CRISPR-based methods, 28 viruses were predicted by BLASTn-short to link with spacers in 10 bins (Supplementary Table 12).

Results show that CRISPR-based results have a very high accuracy. For Illumina data, of the 21 virus-host connections predicted using CRISPR spacers, 11 are simultaneously found in Hi-C data and 10 are not detected in Hi-C data. Of 11 detected connections, only 1 is different in Hi-C data and 10 are the same (91% precision) (Supplementary Table 13). Also for the Nanopore/Illumina hybrid data, of the 28 virus-host connections predicted using CRISPR spacers, 16 are simultaneously found in Hi-C data and 12 are not detected in Hi-C

data. Of 16 detected connections, only 1 is different in Hi-C data, 15 are the same (94% precision) (Supplementary Table 14).

It should be noticed that some of the predicted CRISPR-based virus-host interactions are undetected in Hi-C data. CRISPR spacers represent a collection of memories regarding past virus invasions, while Hi-C data provide a snapshot of ongoing virus-host interactions. Also, Hi-C crosslinking may not be 100% efficient and might miss some of the virus-host interactions.

To supplement CRISPR-based approach, we used Hi-C sequencing data to verify some of the virus-host links inferred by CRISPR spacer assignment and proved the precision of CRISPR-based methods. It has to be noted that Hi-C method also has its limitations, for example, the spurious links due to shared sequence fragments between closely related bacteria or viruses. Overall, a combination of three sets of data in this study, namely targeted metagenomics, direct metagenomics and Hi-C metagenomics, provided a holistic view about the virus-host interactions in complex ecosystems.

Hi-C sequencing, direct Illumina sequencing and Nanopore sequencing

To make further molecular validation about the CRISPR spacer matching results, additional sampling was done in December 2020 at ST WWTP. 100 mL AS samples were first centrifuged at 4500 rpm for 15 mins to remove supernatant. The pellets were split into two parts.

One part of pellets was resuspended with 10 mL 1% formaldehyde to crosslink virus fragments with host fragments and went through 20-min incubation. Formaldehyde-crosslinking was further quenched by glycine. After spinning down pellets, crosslinked AS sample was grinded into a fine powder in a liquid nitrogen-chilled mortar. Hi-C sample was then sent to Phase Genomics (USA) for Hi-C library preparation, DNA sequencing (30 Gb) and proprietary in silico virus-host interaction reconstruction.

The other part was directly extracted for DNA using ZymoBIOMICS DNA Miniprep Kit (Zymo Research, USA). DNA concentrations were measured using a NanoDrop One Spectrophotometer (Thermo Fisher Scientific, USA) and DNA was stored at -20 °C for Illumina and Nanopore sequencing. Illumina sequencing was performed using Illumina Novaseq 6000 PE150 (Novogene, China) for 60 Gb sequencing data. The same DNA also went through in-house Nanopore sequencing which yields 10 Gb sequencing data (Figure 7).

Two strategies have been applied for downstream bioinformatic analysis. For the first strategy, 60 Gb Illumina sequencing data were de novo assembled using CLC Genomics Workbench (version 11.0.1, QIAGEN Bioinformatics, Denmark) with automatic word size and minimum scaffold length of 1 kb. As for the second strategy, 10 Gb Illumina and 10 Gb Nanopore reads were hybrid-assembled using OPERA-MS¹ with pilon polishing. Then, the abovementioned two contigs were each deconvoluted and clustered with 30 Gb Hi-C reads to achieve host genome bins and reconstruct subsequent virus-host interactions.

Figure 7. General workflow to validate the precision of CRISPR-based methods in the present study. For Illumina data, 91% precision was observed. For Nanopore data, 94% precision was observed.

Response to comments of Reviewer 3

Reviewer #3 (Remarks to the Author):

The authors have made good progress towards heeding my suggestions, however certain small issues remain.

Lines 470-472. Contrary to what the response to the review says, there is no mention of having turned on the blast filter for redundant sequences.

Responses: Thanks for the valuable comments.

As for the blast filter, actually we used the `blastn -max_target_seqs` option and set it to be 1. From [NCBI website](https://www.ncbi.nlm.nih.gov/books/NBK279684/table/appendices.T.options_common_to_all_blast/) (https://www.ncbi.nlm.nih.gov/books/NBK279684/table/appendices.T.options_common_to_all_blast/ and <https://ncbiinsights.ncbi.nlm.nih.gov/2019/01/04/blast-2-8-1-with-new-databases-and-better-performance/>), it is said that for version later than BLAST+ 2.8.1, the number of alignments and descriptions will be set to the option `max_target_seqs`. `Max_target_seqs` refers to the maximum number of aligned sequences to keep. We set it to be 1, i.e. turning on the blast filter for redundant sequences.

We have highlighted the usage of option `max_target_seqs` in Line 511-514:

Viral contigs identified for all six WWTPs were searched against manually curated CRISPR-Cas spacer database using `BLASTn-short` to link viruses to their hosts with 97% identity, 90% coverage, 1 mismatch and 1 maximum target sequence (`-max_target_seqs=1`).

Lines 266-231. My comment was about how the biases due to an uneven distribution of CRISPR in prokaryotic groups can affect the results. The abundance of CRISPR spacers doesn't necessarily correlate with how often prokaryotes are attacked by viruses or the number of different viruses attacking the prokaryote. Therefore, I don't think that the phrase "strong potential virus-host interactions" is as adequate, instead there is a higher probability to detect virus-host interactions. If we applied this to particular metabolisms, for some it will

be easier to find corresponding viruses. Yet, it is possible to find many viruses targeting a given metabolism and still have a lot of un-affected strains. The opposite can also be true.

Responses: *Thanks for the valuable comments.*

Following your suggestion, we have revised the conclusions in this paragraph as an uneven distribution of CRISPR in prokaryotic groups.

We have revised this paragraph in the manuscript (Line 235-239):

Notably, for those genera that have more than 10 reference genomes in our curated database, Methanosarcina, Sorangium, Desulfotomaculum and Methanoculleus have more than 100 spacers in their genomes, indicating an uneven distribution of CRISPR in prokaryotic groups (Supplementary Table 8).

Line 461: This is the place to include, about CRT, something as “using the default parameters”.

Responses: *Thanks for the valuable comments.*

Following your suggestion, we have added it in the methods (Line 499-502):

All bacterial and archaeal assembled genomes (N=190,078) from NCBI Assembly (<https://www.ncbi.nlm.nih.gov/assembly>) were retrieved to manually curate a CRISPR-Cas spacer database predicted by CRISPR Recognition Tool (CRT)² using the default parameters.

Line 479: Compare the 0.68% of recall in random sequences to your recall in the datasets for the same real sequences. Is real recall 11-22.6% (line 143)? It could also be stated in line 143 that this implies that the number of erroneous associations can be between 6% ($100 \times 0.68 / 11$) and 3% ($100 \times 0.68 / 22.6$). Also, the process of testing against random sequences is better if more than one replicate is used.

Responses: *Thanks for the valuable comments.*

If we consider 50037 sequences as a whole, $5879 / 50037 = 11.7\%$ should be the real recall rate. Following your suggestion, we have performed recall in random sequences for two more replicates and each of them have $340 / 50037 = 0.68\%$ and $375 / 50037 = 0.75\%$ recall, respectively. We then updated the average random recall rate to be 0.70%. Also, we have added the estimated erroneous percentage 6% ($100 \times 0.7 / 11.7$) in the manuscript.

We have added this part in the revised manuscript (Line 143-147, Line 517-521):

Based on the cutoffs used, we recovered a total of 5,879 viral contigs (4,897 viral genera) (11.7% recall rate) with their predicted hosts, comprising 11.0-22.6% of viral contigs (coverage percentage) in each WWTP. Considering the random recall rate (0.70%) simply happen by chance (see Methods), the percentage of erroneous associations can be 6%.

We have created three replicate databases randomly using RSAT-random sequence (http://rsat.sb-roscoff.fr/random-seq_form.cgi) which each contains the same number (N=50037) and the same size (~614MB) of our total viral sequences. Then we used the same criteria to select the best hit. Results showed that on average 0.70% of DNA sequences have random hits to our curated CRISPR-Cas spacer database.

-Lines 480-481.” However, none of them could link spacers from different domains of life.”

The fact that, with random viral sequences, no domains of life were linked is not representative for the exposed results. As with random sequences the number of associations is expectedly decreased, it is to be expected that the random chance of associating spacers from different domains of life is greatly reduced. For instance, if recall is 22,4% of viruses compared to 0,68%, there is 32 times more opportunities to have an incorrect association to an already paired viral genome in your actual data.

A much adequate approach for that purpose would be to take only the recalled viral genomes, transform them into random sequences, and see recall to the set of spacers from the other domain. It is a good practice to repeat this process a few times, not just one. And then compare to the number of cross-domain pairings obtained. However, this approach has a few pitfalls (viruses that were recalled for two domains and a percentage of the recalls may be incorrect).

Another possibility is to take the estimated percentage of erroneous host identifications (see commentary about line 479). For example, let’s assume an estimate of 3% errors and every virus predicted to infect 2 domains that has two identifications (one for each domain). So the possibility that at least one of these identifications is false equals $P(f)=1-(1-0.03)^2=0,059=5,9\%$; Therefore, of 135 viruses infecting 2 domains of live, about 8.0 (5.9%) would be expected to have happened just by chance.

Responses: *Thanks for the valuable comments.*

Following your suggestion, we estimated the error percentage in host identifications in previous response to be 6%. The possibility that at least one of these identifications is false equals $P(f)=1-(1-0.06)^2=0.1164=11.6\%$. Therefore, of 135 viruses infecting 2 domains of life, about 16 (11.6%) would be expected to have happened just by chance.

We have added these details in the revised manuscript (Line 176 to 180):

Considering the random error percentage 6% in host identifications in our samples, the possibility that at least one of these identifications is false equals $P(f)=1-(1-0.06)^2=0.1164=11.6\%$. Therefore, of 135 viruses infecting 2 domains of life, about 16 (11.6%) would be expected to have happened just by chance.

Reference:

1. Bertrand D, *et al.* Hybrid metagenomic assembly enables high-resolution analysis of resistance determinants and mobile elements in human microbiomes. *Nature Biotechnology* **37**, 937-944 (2019).
2. Bland C, *et al.* CRISPR recognition tool (CRT): a tool for automatic detection of clustered regularly interspaced palindromic repeats. *BMC Bioinformatics* **8**, 209 (2007).

REVIEWERS' COMMENTS

Reviewer #3 (Remarks to the Author):

Most of my comments have been addressed. Also, there has been a miscommunication issue (Line 512). When I mentioned to filter “redundant sequences” I actually meant to turn on the BLAST filter for low complexity regions. But as default BLASTn parameters (-dust) filters them out and, it is unlikely that this will change, it is not something that I care much about now.

However, there are a couple of concerns that I'd like the authors to address.

Line 178. “at least” should be replaced by “each”.

Lines 233-239. There is still the issue of the uneven distribution of CRISPR throughout prokaryotes. This is not a novelty at all. The point is that the number of phage-prey interactions that can be detected using this approach is a minority, and does not always represent present scenarios. Even phages infecting a CRISPR-positive strain may leave no trace in the form of spacers. Therefore, the conclusions that derive from these predictions must be handled with caution. Phages are most probably affecting all organisms and metabolisms, and it would be surprising if they weren't. So, what I would like the authors to do is, to include a disclaimer about these limitations of their method. Something similar to what they do about Hi-C in lines 322 to 326.

Reviewer #4 (Remarks to the Author):

Authors have addressed concerns previously raised by reviewer 2 including additional experiments to substantiate their hypothesis. Additionally, limitations of their study have been included in the discussion section.

Point-by-point response to the reviewers' comments

Manuscript ID: NCOMMS-20-23333C-Z

Title: Prokaryotic viruses impact functional microorganisms in nutrient removal and carbon cycle in wastewater treatment plants

Authors: Yiqiang Chen, Yulin Wang, David Paez-Espino, Martin F. Polz, Tong Zhang

Corresponding author: Prof. Tong Zhang

We highly appreciate the valuable comments from reviewers. We have revised our manuscript according to these comments. The revisions have been highlighted with yellow color. The point-by-point responses are presented as follows.

Response to comments of Reviewer 3

Reviewer #3 (Remarks to the Author):

Most of my comments have been addressed. Also, there has been a miscommunication issue (Line 512). When I mentioned to filter “redundant sequences” I actually meant to turn on the BLAST filter for low complexity regions. But as default BLASTn parameters (-dust) filters them out and, it is unlikely that this will change, it is not something that I care much about now.

Responses: *Thanks for the positive comments and your efforts in handling this manuscript.*

However, there are a couple of concerns that I'd like the authors to address.

Line 178. “at least” should be replaced by “each”.

Responses: *Thanks for the valuable comments.*

Following your suggestion, we have revised it in Line 178:

each one of these identifications is false equals $P(f)=1-(1-0.06)^2=0.1164=11.6\%$.

Lines 233-239. There is still the issue of the uneven distribution of CRISPR throughout prokaryotes. This is not a novelty at all. The point is that the number of phage-prey

interactions that can be detected using this approach is a minority, and does not always represent present scenarios. Even phages infecting a CRISPR-positive strain may leave no trace in the form of spacers. Therefore, the conclusions that derive from these predictions must be handled with caution. Phages are most probably affecting all organisms and metabolisms, and it would be surprising if they weren't. So, what I would like the authors to do is, to include a disclaimer about these limitations of their method. Something similar to what they do about Hi-C in lines 322 to 326.

Responses: Thanks for the valuable comments.

Following your suggestion, we have added the limitation part in the manuscript (Line 234-240):

Results showed that these Midas genera contain on average 44 spacers in their genome and there exists an uneven distribution of CRISPR in prokaryotic groups (Supplementary Table 8). It should be noticed that the number of phage-prey interactions that can be detected using this approach is a minority and does not always represent present scenarios. Even phages infecting a CRISPR-positive strain may leave no trace in the form of spacers. Therefore, the conclusions that derive from these predictions must be handled with caution.

Response to comments of Reviewer 4

Reviewer #4 (Remarks to the Author):

Authors have addressed concerns previously raised by reviewer 2 including additional experiments to substantiate their hypothesis. Additionally, limitations of their study have been included in the discussion section.

Responses: Thanks for the positive comments and your efforts in handling this manuscript.